# Language Models as Implicit Tree Search

**Ziliang Chen** [1]  **Zhao-Rong Lai** [2]  **Yufeng Yang** [3]  **Liangda Fang** [2]  **Zhanfu Yang** [4]  **Liang Lin** [1 3]

## Abstract

Despite advancing language model (LM) alignment, direct preference optimization (DPO) falls short in LM reasoning with the free lunch from reinforcement learning (RL). As the breakthrough, this work proposes a new RL-free preference optimization method aiming to achieve DPO along with learning another LM, whose response generation policy holds the asymptotic equivalence with AlphaZero-like search, the apex of algorithms for complex reasoning missions like chess Go. While circumventing explicit value and reward modeling, the neural implicit tree search executed by the extra LM remains seeking to equip DPO with reasoning procedure technically akin to AlphaZero. Our experiments demonstrate that our methodology outperforms both regular DPO variants in human preference alignment, and MCTS-based LMs in mathematical reasoning and planning tasks.

## 1. Introduction

Preference optimization paradigms, notably Reinforcement Learning from Human Feedback (RLHF) (Bai et al., 2022) and Direct Preference Optimization (DPO) (Rafailov et al., 2024b), are foundational to modern Language Model (LM) alignment, aiming to imbue these models with human's behavioral characteristics, inclination, and value. In particular, DPO circumvent reward modeling so as to directly optimize LMs with preferential response pairs, thereby reinforcement learning (RL)'s complexities and instability avoided to yield more efficiently and stably aligned models.

Distinct from preference alignment, sophisticated reasoning capabilities underlying human brains are not typically replicated by RLHF and DPO alone: the emulation often necessitates synergistic integration with other techniques, *i.e.*, meticulous prompt engineering (Wei et al., 2022), process-based reward modeling (Lightman et al., 2023), and Monte Carlo Tree Search (MCTS) (Xie et al., 2024a). Among these complementary approaches, MCTS stands out as essentially promising, due to its exploration of complex decision spaces in achieving high-level reasoning tasks such as game Go (Silver et al., 2017b). Massive leading research were proposed to combine MCTS with preference optimization, whereas their MCTS procedures inevitably require the value function learned by RL for executing the search principles of Upper Confidence Bound (UCB). It is noteworthy that the critical advantage of DPO is to skip the value learning to achieve faster and more stable preference optimization than RLHF. The standard MCTS's reliance on an RL-learned value function directly conflicts with DPO's core strength of bypassing RL entirely, showing a significant hurdle for their seamless integration. It implies the necessity to reconcile MCTS methodologies with the RL-free nature behind DPO.

Rather than the regular MCTS, this paper started from its theoretical variant derived from (Grill et al., 2020b), where AlphaZero-like MCTS can be treated as a stochastic policy solved by the state-specific local optimization regularized by the reverse KL-divergence. With this regard, each state-specific local tree search decision asymtotically converges to AlphaZero-like MCTS decision with the gap bounded by the empirical visit counts, thus, lifting the expressiveness of MCTS limited by the integer-count probability and sparse tree width. Regretfully, the local solution of this stochastic policy involves the parameter only solvable by dichotomic search per state. In terms of the exponentially complex state space in token-level language generation, this *implicit tree search* (ITS) is hardly applied in MCTS-based LM research.

**Contributions.** We propose a new RL-free preference optimization paradigm aiming to approximate ITS by LMs. Specifically,

- Beyond the original token-selection policy in DPO, we propose another LM to learn the global policy of ITS with regards to neural universal approximation. The preference optimization built on top of DPO, without dichotomic search and any other RL elements, inherits the critical advantages of the DPO family.

---

[1]Research Institute of Multiple Agents and Embodied Intelligence, Peng Cheng Laboratory; [2]Jinan University, Guangzhou, China; [3]School of Computer Science and Engineering, Sun Yatsen University, Guangzhou, China; [4]Department of Computer Science, Rutgers University. Correspondence to: Liang Lin <linlng@mail.sysu.edu.cn>.

*Proceedings of the 42$^{nd}$ International Conference on Machine Learning*, Vancouver, Canada. PMLR 267, 2025. Copyright 2025 by the author(s).

- In terms of the gradient analysis to the preference optimization with ITS, learning its global policy yields the response generation with diverse and better aligned preference, therefore, simultaneously benefits AI alignment and reasoning tasks.

- Self-improved preference policy augmentation and decoding strategy are proposed for our ITS-based preference optimization approaches, in order to align with the MCTS-based LLM decoding. The connection between the decoding strategies and group-relative policy optimization (Shao et al., 2024) has also been presented.

Our experiments included the evaluation across human preference alignment, mathematical reasoning, and mathematical planning, where our approach concurrently reaped the optima against DPO variants and MCTS-derived baselines.

## 2. Preliminaries

For language generation, language models (LM) $\theta$ serve as a token-level Markov decision process (MDP) $(\mathcal{S},\mathcal{A},\mathcal{P},\mathcal{R},\mathcal{T})$, where the state $s_t = (x, y_{<t})$ in $\mathcal{S}$ consists of a prompt $x$ followed by a sequence of response tokens $x_{<t}$ generated at the previous $t$-1 steps, and the action $a_t = y_t$ in $\mathcal{A}$ denotes the token selected at the current $t$ step. Given this, the transition kernel $\mathcal{P}: (\mathcal{S},\mathcal{A}) \rightarrow \mathcal{S}$ holds the deterministic mapping from $(s_t, a_t)$ to the next state $s_{t+1} = (x, y_{<t+1})$ with $y_{<t+1} = (y_{<t}, y_t)$. The reward $\mathcal{R}: (\mathcal{S},\mathcal{A}) \rightarrow \mathbb{R}$ quantifies the preference of selecting tokens, and $T \in \mathcal{T}$ denotes the step to cease the generation when $a_T$ is the end-of-sequence (**EoS**) token. The goal is to learn the sequential token-selection policy $\pi_\theta$ that maximizes the accumulated reward $R(x,y) = \sum_{t=1}^{T} r(s_t, a_t), \forall r \in \mathcal{R}$.

**RLHF and DPO**. Token-level MDP needs $R(x, y)$ to reflects human preferences. RLHF (Bai et al., 2022) captures this from prompt $x$ with its response pairs $y^{(w)} \succ y^{(l)}$, where $\succ$ identifies $y^{(w)}$ more preferable than $y^{(l)}$. The comparison is made by Bradley-Terry (BT) model (Bradley & Terry, 1952) in the contextual bandit setting

$$P_{\text{BT}}(y^{(w)} \succ y^{(l)}|x) = \frac{\exp(R(x, y^{(w)}))}{\exp(R(x, y^{(w)})) + \exp(R(x, y^{(l)}))}.$$
(1)

where $R(x, y)$ can be learned through

$$\min_R -\mathbb{E}_{(x,y^{(w)},y^{(l)}) \sim D}[\log \sigma(R(x, y^{(w)}) - R(x, y^{(l)}))]$$
(2)

with the human preference dataset $D$. It serves RLHF as a KL-constraint RL objective

$$\max_{\pi_\theta} \mathbb{E}_{x \sim \mathcal{D}, y \sim \pi_\theta(\cdot|x)}[R(x, y) - \beta D_{KL}(\pi_\theta(\cdot|x) \| \pi_{\text{ref}}(\cdot|x))],$$
(3)

where the reference policy $\pi_{\text{ref}}$ is initialized by supervised fine-tuning (SFT) LM to learn the generation policy $\pi_\theta$ via PPO (Schulman et al., 2017), and $\beta > 0$ controls their

trade off. DPO (Rafailov et al., 2024b) reconsiders the optimization from a interpretation of $R(x, y)$ represented as $R(x, y) = \beta \log \frac{\pi_\theta(y|x)}{\pi_{\text{ref}}(y|x)} + \beta \log Z(x)$. With such regards, Eq.2 is equivalent with

$$\mathcal{L}_{\text{DPO}}(\pi_\theta; \pi_{\text{ref}}) = -\mathbb{E}_{(x,y^{(w)},y^{(l)}) \sim \mathcal{D}}$$
$$\left[ \log \sigma \left( \beta \log \frac{\pi_\theta(y_w|x)}{\pi_{\text{ref}}(y_w|x)} - \beta \log \frac{\pi_\theta(y_l|x)}{\pi_{\text{ref}}(y_l|x)} \right) \right],$$
(4)

therefore the reward modeling phase is skipped to directly optimize the token-selection policy $\pi_\theta$.

**Token-level Interpretation of LM Alignments.** The aforementioned LM alignments focus on a complete response $y$, yet $\pi_\theta$ is executed in each MDP step using token-level rewards $r(s_t, a_t)$ with $\beta \log \frac{\pi_\theta(y_t|[x,y_{<t}])}{\pi_{\text{ref}}(y_t|[x,y_{<t}])} = \beta \log \frac{\pi_\theta(a_t|s_t)}{\pi_{\text{ref}}(a_t|s_t)}$. (Rafailov et al., 2024a) re-formulates Eq.3 via the token-level interpretation

$$\max_{\pi_\theta} \mathbb{E}_{s_0=x \sim \mathcal{D}, a_t \sim \pi_\theta(\cdot|s_t)} \sum_{t=1}^{T_y} \left[ r(s_t, a_t) - \beta \log \frac{\pi_\theta(a_t|s_t)}{\pi_{\text{ref}}(a_t|s_t)} \right]$$
$$\leftrightarrows \max_{\pi_\theta} \mathbb{E}_{s_0=x \sim \mathcal{D}, a_t \sim \pi_\theta(\cdot|s_t)} \sum_{t=1}^{T_y} \left[ r(s_t, a_t) + \beta \log \pi_{\text{ref}}(a_t|s_t) + \beta \mathcal{H}(\pi) \right].$$
(5)

So given $r_{\text{ME}}(s_t, a_t) = r(s_t, a_t) + \beta \log \pi_{\text{ref}}(a_t|s_t)$, RLHF with the reframed reward is maximum-entropy RL (Ziebart, 2010) whose fixed point solution leads to the optimal policy $\pi_\theta^*$ that satisfies $\pi_\theta^*(a_t|s_t) = \exp\left(\frac{Q^*(s_t, a_t) - V^*(s_t)}{\beta}\right)$ with respect to $V^*(s_t) = \beta \log \sum_{a \in \mathcal{A}} \exp\left(Q^*(s_t, a_t)/\beta\right)$. It is noteworthy that the optimal state-action value function holds the initial value $Q^*(s_0, a_0) = R(x, y) = V^*(s_0) + \beta \sum_{t=1}^{T_y} \log \frac{\pi_\theta^*(a_t|s_t)}{\pi_{\text{ref}}(a_t|s_t)}$ with $V^*(s_0) = V^*(x)$, henceforth we have the objective similarly derived from DPO

$$\mathcal{L}_{\text{T-DPO}}(\pi_\theta; \pi_{\text{ref}}) = -\mathbb{E}_{(x,y^{(w)},y^{(l)}) \sim \mathcal{D}}$$
$$\left[ \log \sigma \left( \beta \sum_{t=1}^{T_w} \log \frac{\pi_\theta(a_t^w|s_t^w)}{\pi_{\text{ref}}(a_t^w|s_t^w)} - \beta \sum_{t=1}^{T_l} \log \frac{\pi_\theta(a_t^l|s_t^l)}{\pi_{\text{ref}}(a_t^l|s_t^l)} \right) \right],$$
(6)

where $a_t^w$, $a_t^l$ indicate the $t$-th token of the responses $y^w$, $y^l$; and $s_t^w = [x, y_{<t}^w]$, $s_t^l = [x, y_{<t}^l]$, respectively.

**MCTS and AlphaZero-like Tree Search.** Given the state-action value treated as feedback from a contextual bandit, MCTS (Browne et al., 2012) builds an online search tree incrementally for decision making. (More details refers to (Silver et al., 2017b)).MCTS algorithms are famous in massive researches, among which the most successful case is Alpha Zero (Silver et al., 2017b) and its variants (Silver et al., 2017b; Schrittwieser et al., 2020). The MCTS variant underlying these methods remarked as *AlphaZero-like tree search*, compute a policy at the root of the search tree improved from the distribution predicted by the prior policy

$\pi$, which is updated by distilling back the tree-search policy to update the prior. The significant difference from other MCTS is AlphaZero-like tree incorporating $\pi$ as the prior during the search procedure. It leads to the heuristic search principle inspired from UCB:

$$\boldsymbol{a}^{\star} \triangleq \underset{\boldsymbol{a} \in \mathcal{A}_{\mathsf{MCTS}}}{\arg\max} \left[ Q^{\pi}(\boldsymbol{s}, \boldsymbol{a}) + c\pi(\boldsymbol{a}|\boldsymbol{s}) \frac{\sqrt{\sum\limits_{\boldsymbol{a}' \in \mathcal{A}_{\mathsf{MCTS}}} n(\boldsymbol{s}, \boldsymbol{a}')}}{1 + n(\boldsymbol{s}, \boldsymbol{a})} \right]. \tag{7}$$

where $c > 0$ is a balance factor between exploration and exploitation in the MCTS search space $\mathcal{A}_{\mathsf{MCTS}}$. The four-stage procedure of Eq.7 is shown in Appendix.A.

## 3. Implicit Tree Search: Warm-Up

AlphaZero-like MCTS variants have been intensively explored in leading researches for LM training (Feng et al., 2023; Zhang et al., 2024a) and alignment (Xie et al., 2024b; Chen et al., 2024a), in order to mitigate their downsides of reasoning and long-form generation ability captured in post-training. In spite of different algorithm designs, the methodologies are consistent with MCTS in the data-generation manner: $\hat{\pi}$ merges with the decoding strategy to generate responses for training data augmentation, then the token-selection policy $\pi_{\theta}$ and visit counts updated along with these data, further renew the decoding by Eq.7. Despite significantly advancing LM alignments, the decision-making pipeline relies on the online state-action visit count $n(\boldsymbol{s}, \boldsymbol{a})$ per state-action pair. In terms of the integer-count probability and the sparse tree width in the initial training iters, it inevitably suffers from two critical problems.

**Cold-start exploration expressiveness**. $\frac{\sqrt{\sum_{\boldsymbol{a}' \in \mathcal{A}} n(\boldsymbol{s}, \boldsymbol{a}')}}{1 + n(\boldsymbol{s}, \boldsymbol{a})}$ is the key to differentiate $\hat{\pi}$ and $\pi$, so AlphaZero-like tree search typically employs its *empirical visit distribution* $\frac{1 + n(\boldsymbol{s}, \boldsymbol{a})}{|\mathcal{A}_{\mathsf{MCTS}}| + \sum_{\boldsymbol{a}' \in \mathcal{A}_{\mathsf{MCTS}}} n(\boldsymbol{s}, \boldsymbol{a}')}$ or its exponential generalization $\frac{1 + (n(\boldsymbol{s}, \boldsymbol{a}'))^r}{|\mathcal{A}_{\mathsf{MCTS}}| + \sum_{\boldsymbol{a}' \in \mathcal{A}_{\mathsf{MCTS}}} (n(\boldsymbol{s}, \boldsymbol{a}'))^r}$ to explore the tree structure. Whereas the discrete nature of visit counts limit the expressiveness of exploration strategy.

**Proposition 3.1.** *Given a fixed $r \in \mathbb{R}/\{0\}$, the action selection probability by the exponential empirical visit distribution holds its value with the equal cardinality of $\mathbb{Z}$.*

**Proposition 3.2.** *Given a learnable $r \in \mathbb{R}$, the action selection probability by the exponential empirical visit distribution holds the value in the range $(\frac{1}{|\mathcal{A}_{\mathsf{MCTS}}|}, 1)$.*

The problem is cold-start because early decisions heavily influences the tree growth yet the ratios of integers are more unstable when the visit counts are small. It may drive the language generation into bias or sub-optima.

**Sparse-action improvement**. $\hat{\pi}$ solely improved for actions

searched at least once by the previous simulation whereas those with $n(\boldsymbol{s}, \boldsymbol{a})=0$ would be dominated by $\theta$. In terms of the large action space in language and the deterministic policy in Eq.7, it may cause a large simulation budget to improve actions never visited.

Recognizing that these limitations are intrinsic to the use of online visit counts in conventional MCTS algorithms, we supersedes the online tree-search strategy with stochastic policy optimization established in (Grill et al., 2020b) (depicted in Figure.1.b), named *Implicit Tree Search* since the optimized policy is provably equivalent with the AlphaZero-like tree search exploration defined in Eq.7.

### 3.1. MCTS as Regularized Policy Optimization

More specifically, for any state, the action selection formula in Eq.7 holds the identical interpretation

$$\boldsymbol{a}^{\star}(\boldsymbol{s}) \triangleq \underset{\boldsymbol{a} \in \mathcal{A}_{\mathsf{MCTS}}}{\arg\max} \left[ Q^{\pi}(\boldsymbol{s}, \boldsymbol{a}) + \lambda_N(\boldsymbol{s}) \cdot \frac{\pi(\boldsymbol{a}|\boldsymbol{s})}{\hat{\pi}(\boldsymbol{a}|\boldsymbol{s})} \right] \tag{8}$$

where $\hat{\pi}(\boldsymbol{a}|\boldsymbol{s}) \triangleq \frac{1 + n(\boldsymbol{s}, \boldsymbol{a})}{|\mathcal{A}_{\mathsf{MCTS}}| + \sum_{\boldsymbol{a}' \in \mathcal{A}_{\mathsf{MCTS}}} n(\boldsymbol{s}, \boldsymbol{a}')}$ represents the empirical visit distribution and $\lambda_N(\boldsymbol{s}) \triangleq c \cdot \frac{\sqrt{\sum_{\boldsymbol{a}' \in \mathcal{A}_{\mathsf{MCTS}}} n(\boldsymbol{s}, \boldsymbol{a}')}}{|\mathcal{A}_{\mathsf{MCTS}}| + \sum_{\boldsymbol{a}' \in \mathcal{A}_{\mathsf{MCTS}}} n(\boldsymbol{s}, \boldsymbol{a}')}$ denotes the *state-specific multiplier*. Notice that for any $\boldsymbol{s} \in \mathcal{S}$, the empirical visit distribution $\hat{\pi}(\cdot|\boldsymbol{s})$ is the only way that search algorithm influences the optimal action of tree search. In language generation context, we set $\pi = \pi_{\theta}$, so for any $\boldsymbol{s} \in \mathcal{S}$, $\hat{\pi}(\cdot|\boldsymbol{s})$ holds a corresponding predominant policy $\overline{\pi}(\cdot|\boldsymbol{s})$ as

**Theorem 3.3.** *(Asymptotic equivalence between ITS and MCTS policies) $\forall \boldsymbol{s} \in \mathcal{S}$, let $\overline{\pi}(\cdot|\boldsymbol{s})$ be the solution of*

$$\begin{aligned} \overline{\pi}(\cdot|\boldsymbol{s}) \triangleq \underset{\mathbf{y}(\boldsymbol{s}) \in \mathbb{S}}{\arg\max} \Big[ & \mathbf{Q}^{\pi_{\theta}}(\boldsymbol{s})^{\top} \mathbf{y}(\boldsymbol{s}) \\ & - \lambda_N(\boldsymbol{s}) D_{KL}[\pi_{\theta}(\cdot|\boldsymbol{s}), \mathbf{y}(\boldsymbol{s})] \Big] \end{aligned} \tag{9}$$

*where $\mathbb{S}$ denotes the $|\mathcal{A}_{\mathsf{MCTS}}|$-dimensional simplex, and $\mathbf{Q}^{\pi_{\theta}}(\boldsymbol{s}) = \big( Q^{\pi_{\theta}}(\boldsymbol{s}, \boldsymbol{a}_1), Q^{\pi_{\theta}}(\boldsymbol{s}, \boldsymbol{a}_2), \cdots, Q^{\pi_{\theta}}(\boldsymbol{s}, \boldsymbol{a}_{|\mathcal{A}_{\mathsf{MCTS}}|}) \big)$ is the Q-value vector with respect to the state $\boldsymbol{s}$. Then as the visit counts increase, the empirical visit distribution $\hat{\pi}(\cdot|\boldsymbol{s})$ in Eq.8 converges to $\overline{\pi}(\cdot|\boldsymbol{s})$ with the upper bound*

$$\left\| \hat{\pi}(\cdot|\boldsymbol{s}) - \overline{\pi}(\cdot|\boldsymbol{s}) \right\| \leq \frac{|\mathcal{A}_{\mathsf{MCTS}}| + 1}{|\mathcal{A}_{\mathsf{MCTS}}| + N} \tag{10}$$

*where $N$ indicates the total rounds of simulation.*

Theorem.3.3 is concluded from the definition.1 and proposition.1,5 in (Grill et al., 2020b). The theoretical result verifies that, given sufficient simulation steps, there exists an asymptotic equivalence between the exploration of AlphaZero-like tree search $\hat{\pi}(\cdot|\boldsymbol{s})$ and the stochastic-sampling policy $\overline{\pi}(\cdot|\boldsymbol{s})$. Note that the policy optimization in a simplex allows the

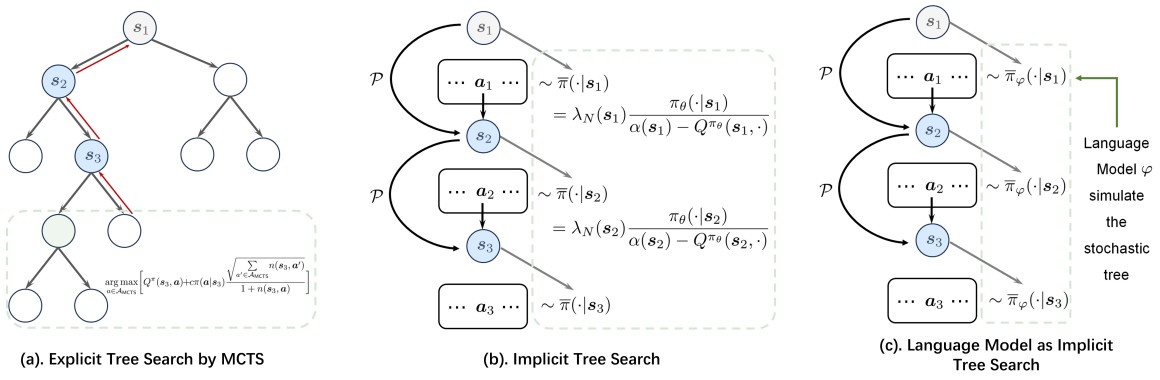

Figure 1. Comparison between the diagrams of (a).MCTS (regular MCTS and AlphaZero-like tree search), (b). Implicit Tree Search (ITS) (Grill et al., 2020b), and our IT-PO algorithm.

continuous value change in $\overline{\pi}(\cdot|s)$ that prevents the risk of cold-start expressiveness. Besides, the reversed KL divergence is smooth on $\overline{\pi}(\cdot|s)$, ensuring the sparse-action improvement resolved.

Accordingly, the ***expand*** and ***backup*** stages along with the MCTS procedure require the state-specific stochastic policy optimization formulated as follows

**Lemma 3.4.** *(Solution of ITS policy (Grill et al., 2020b))* $\forall s \in \mathcal{S}$, the solution $\overline{\pi}(\cdot|s)$ of Eq.9 holds

$$\forall a_t \in \mathcal{A}_{\text{MCTS}}, \ \ \overline{\pi}(a_t|s_t) = \lambda_N(s_t)\frac{\pi_\theta(a_t|s_t)}{\alpha(s_t) - Q^{\pi_\theta}(s_t, a_t)} \tag{11}$$

*where $\alpha(s_t)$ is defined as*

$$1).\alpha(s_t) \triangleq \max\left\{\alpha \in \mathbb{R}, \text{ s.t. } \sum_{a \in \mathcal{A}_{\text{MCTS}}} \pi(a|s_t) = 1\right\}$$

$$2).\alpha(s_t) \geq \alpha(s_t)_{\min} \triangleq \max_{a \in \mathcal{A}_{\text{MCTS}}}\left(Q(s_t, a) + \lambda_N\pi_\theta(a|s_t)\right)$$

$$3).\alpha(s_t) \leq \alpha(s_t)_{\max} \triangleq \max_{a \in \mathcal{A}_{\text{MCTS}}} Q(s_t, a) + \lambda_N \tag{12}$$

Lemma.3.4 is derived from Appendix.B.3 in the paper.

### 3.2. Challenges of Combing Implicit Tree and LMs

Observed that $\sum_{a \in \mathcal{A}_{\text{MCTS}}}\pi(a|s_t)$ monotonically decreases on $(\alpha(s_t)_{\min}, \alpha(s_t)_{\max})$, the theoretical result guarantees the state-specific hyper-parameter $\alpha(s_t)$ uniquely identified using dichotomic search over $(\alpha(s_t)_{\min}, \alpha(s_t)_{\max})$. In other words, $\overline{\pi}$ **can not be universally optimized by gradient descent and even for each state, $\overline{\pi}(\cdot|s_t)$'s solution in Eq.11 is not exactly closed-form**. To this end, $\overline{\pi}$ can not be flexibly used as explicit search like regular MCTS algorithms and instead, only applicable for policy distillation to update the $Q$ function in AlphaZero-like MCTS. However, it is hardly applied for LLM-based preference optimization.

The vital problem roots in the exponentially increase size of the valid state space. As demonstrated in token-level MDP, texts are generated by sequential token selection with the state $s_t = (x, y_{<t})$ deterministically transmitted as $\mathcal{P}(s_t, a_t) \to s_{t+1} = (x, y_{<t+1})$. It implies that for each prompt $x$, the valid state space holds the size $|V|^t$ ($|V|$ indicates the size of token vocabulary $V$) exponentially increase with $t$. Given such challenge, Lemma.3.4 solely promises the local policy so that the actor-critic modeling is required to obtain $\pi_\theta$ and $Q^{\pi_\theta}$ simultaneously, as leading LLM-based MCTS approaches do. What's worse, the local policy also relies on $\alpha(s_t)$ inferred by dichotomic search per state, implying the computation and buffer also exponentially increasing. This problem even more severe when the action space $\mathcal{A}_{\text{MCTS}}$ become sentence-level.

## 4. Preference Optimization with Implicit Tree Search

In the previous section, we demonstrated the advantages of $\overline{\pi}$ over the empirical visit exploration employed by regular MCTS methods, while its solution of Lemma.3.4 is hardly implemented in the LM realm due to the state-specific non-differentiable hyper-parameters $\alpha(s_t)$. In this section, we derive the new RL-free preference optimization paradigm to approximate $\overline{\pi}$ without either $Q$ function or $\alpha(s_t)$, yielding the response generation as MCTS without value modeling.

### 4.1. Language Models as ITS Policy

Specifically, we chase for the universal approximator of $\overline{\pi}$ based on the extra policy network $\overline{\pi}_\varphi$ parameterized by the other language model $\varphi$ (Figure.1.c). It supports the instant inference of $\overline{\pi}(\cdot|s_t)$ (*i.e.*, no dichotomic search required to obtain $\alpha(s_t)$), no matter $s_t$ was visited or not.

Thought consistent with $\pi_\theta$ in the reversed KL constraint, $\overline{\pi}$ is essentially a stochastic strategy generalized from deterministic policy in the AlphaZero-like tree search (Theo-

rem.3.3). It motivates us to initialize its universal approximator $\overline{\pi}_\varphi$ by supervised fine-tuning. Specifically, we fine-tune a pre-trained LM to obtain $\pi_{\text{sft}}$, whose parameters initialize the reference policy $\pi_{\text{ref}}$, the token selection policy $\pi_\theta$, and our ITS policy approximator $\overline{\pi}_\varphi$. Then we applied the DPO variant algorithm, *i.e.*, conservative DPO (cDPO) (Mitchell, 2023b) to update $\pi_\theta$:

$$\mathcal{L}_{\text{cDPO}}(\theta) = \mathbb{E}_{(\boldsymbol{x}, \boldsymbol{y}^{(w)}, \boldsymbol{y}^{(l)}) \sim D}$$
$$-[(1-\epsilon)\log[\sigma(u_\theta(\boldsymbol{x}, \boldsymbol{y}^{(w)}, \boldsymbol{y}^{(l)}))]+\epsilon\log[\sigma(u_\theta(\boldsymbol{x}, \boldsymbol{y}^{(l)}, \boldsymbol{y}^{(w)}))]$$
$$(13)$$

where $u(\cdot)$ indicates the preference logit derived from BT model, $\epsilon \in [0, 0.2)$ denotes the margin then when $\epsilon = 0$, cDPO degenerates into DPO.

After training with $\min_\varphi \mathcal{L}_{\text{cDPO}}(\varphi; \theta)$, the parameters of $\pi_\theta$, $\pi_{\text{ref}}$ are frozen to update the parameter of $\overline{\pi}_\varphi$. This manner provides the stable $\pi_\theta$ to derive the optimization of $\varphi$, *i.e.*, Implicit-Tree Preference Optimization (IT-PO).

### 4.2. Implicit Tree Preference Optimization (IT-PO)

IT-PO seeks for learning $\overline{\pi}_\varphi$ to substitute $\overline{\pi}$ without either reward or value functions. It is noteworthy that $\overline{\pi}(\cdot|\boldsymbol{s}_t)$ inherently contains $Q^{\pi_\theta}(\cdot, \boldsymbol{s}_t)$ from Lemma.3.4, which serves as the implicit policy evaluation for $\overline{\pi}$ with the lower values implying increased exploration and vice versa. Therefore if $\overline{\pi}_\varphi$ approximate $\overline{\pi}$ well enough, ITS can implicitly derive the value function from $\overline{\pi}_\varphi$ without explicit modeling $Q^{\pi_\theta}$.

For simplicity, we first consider step-synchronous ITS with the policy $\pi_\theta$, wherein the MCTS action node is token-level, *i.e.*, $\mathcal{A}_{\text{MCTS}} = \mathcal{A}$. Specifically, given fixed $\theta$ and rounds of simulation (*i.e.*, $\lambda_N(\boldsymbol{s}_t)$ is also fixed), the ITS policy $\overline{\pi}(\cdot|\boldsymbol{s}_t), \forall \boldsymbol{s}_t \in \mathcal{S}$ can be uniquely determined according to Lemma.3.4. Suppose that $\varphi$ achieve the optimal parameter such that $\overline{\pi}_\varphi(\boldsymbol{a}_t|\boldsymbol{s}_t) = \overline{\pi}(\boldsymbol{a}_t|\boldsymbol{s}_t)$, it holds the observation

**Lemma 4.1.** *Suppose* $(\boldsymbol{x}, \boldsymbol{y}^{(w)})$ *denotes a pair of prompt and its preferred response. For each state* $\boldsymbol{s}_t^{(w)} = (\boldsymbol{x}, \boldsymbol{y}_{<t}^{(w)})$, *the action selects either the preferred token* $\boldsymbol{a}_t^{(w)} = y_t^{(w)}$ *or a random token* $\hat{\boldsymbol{a}}_t$ *sampled from* $\overline{\pi}_{\varphi^*}(\cdot|\boldsymbol{s}_t^{(w)})$*; and each state* $\boldsymbol{s}_t^{(l)} = (\boldsymbol{x}, \boldsymbol{y}_{<t}^{(l)})$, *the action selects either the dispreferred token* $\boldsymbol{a}_t^{(l)} = y_t^{(l)}$ *or a random token* $\check{\boldsymbol{a}}_t$ *sampled from* $\overline{\pi}_{\varphi^*}(\cdot|\boldsymbol{s}_t^{(l)})$. *It holds*

$$R(\boldsymbol{x}, \boldsymbol{y}^{(w)}) =$$
$$\sum_{t=1}^{T_w}\left(r(\boldsymbol{s}_t^{(w)}, \hat{\boldsymbol{a}}_t) - \lambda_N(\boldsymbol{s}_t^{(w)})\left(\frac{\pi_\theta(\boldsymbol{a}_t^{(w)}|\boldsymbol{s}_t^{(w)})}{\overline{\pi}_\varphi(\boldsymbol{a}_t^{(w)}|\boldsymbol{s}_t^{(w)})} - \frac{\pi_\theta(\hat{\boldsymbol{a}}_t|\boldsymbol{s}_t^{(w)})}{\overline{\pi}_\varphi(\hat{\boldsymbol{a}}_t|\boldsymbol{s}_t^{(w)})}\right)\right);$$
$$R(\boldsymbol{x}, \boldsymbol{y}^{(l)}) =$$
$$\sum_{t=1}^{T_l}\left(r(\boldsymbol{s}_t^{(l)}, \check{\boldsymbol{a}}_t) - \lambda_N(\boldsymbol{s}_t^{(l)})\left(\frac{\pi_\theta(\boldsymbol{a}_t^{(l)}|\boldsymbol{s}_t^{(l)})}{\overline{\pi}_\varphi(\boldsymbol{a}_t^{(l)}|\boldsymbol{s}_t^{(l)})} - \frac{\pi_\theta(\check{\boldsymbol{a}}_t|\boldsymbol{s}_t^{(l)})}{\overline{\pi}_\varphi(\check{\boldsymbol{a}}_t|\boldsymbol{s}_t^{(l)})}\right)\right).$$
$$(14)$$

The lemma demonstrates that we may construct the accumulated reward of preference prompt-response pair by the ITS exploration starts from each state of the preferred and dispreferred sequences (*i.e.*, root or intermediate node). As demonstrated in (Rafailov et al., 2024a), DPO-like methods hold the dense reward re-parameterized as $r(\boldsymbol{s}_t, \boldsymbol{a}_t) = \beta \log \pi_\theta(\boldsymbol{a}_t|\boldsymbol{s}_t) - \beta \log \pi_{\text{ref}}(\boldsymbol{a}_t|\boldsymbol{s}_t)$. Combine this with Lemma.4.1 leading to

**Theorem 4.2.** *(Step-Synchronous IT-PO) A prompt* $\boldsymbol{x}$ *drawn from* $D$ *has a response pairs* $\boldsymbol{y}^{(w)} \succ \boldsymbol{y}^{(l)}$. *Given* $\hat{\boldsymbol{A}} = \{\hat{\boldsymbol{a}}_t \sim \overline{\pi}_\varphi(\cdot|\boldsymbol{s}_t^{(w)})\}_{t=1}^{T_w}$ *and* $\check{\boldsymbol{A}} = \{\check{\boldsymbol{a}}_t \sim \overline{\pi}_\varphi(\cdot|\boldsymbol{s}_t^{(l)})\}_{t=1}^{T_l}$, *it holds*

$$U_{\text{ss}}(\boldsymbol{x}, \boldsymbol{y}^{(w)}, \boldsymbol{y}^{(l)}) = R(\boldsymbol{x}, \boldsymbol{y}^{(w)}) - R(\boldsymbol{x}, \boldsymbol{y}^{(l)})$$
$$= \mu_w(\varphi, \theta) - \mu_l(\varphi, \theta) + \delta(\theta),$$

$$\text{s.t. } \mu_w(\varphi, \theta) = -\sum_{t=1}^{T_w}\lambda_N(\boldsymbol{s}_t^{(w)})\left(\frac{\pi_\theta(\boldsymbol{a}_t^{(w)}|\boldsymbol{s}_t^{(w)})}{\overline{\pi}_\varphi(\boldsymbol{a}_t^{(w)}|\boldsymbol{s}_t^{(w)})} - \frac{\pi_\theta(\hat{\boldsymbol{a}}_t|\boldsymbol{s}_t^{(w)})}{\overline{\pi}_\varphi(\hat{\boldsymbol{a}}_t|\boldsymbol{s}_t^{(w)})}\right)$$

$$\mu_l(\varphi, \theta) = -\sum_{t=1}^{T_l}\lambda_N(\boldsymbol{s}_t^{(l)})\left(\frac{\pi_\theta(\boldsymbol{a}_t^{(l)}|\boldsymbol{s}_t^{(l)})}{\overline{\pi}_\varphi(\boldsymbol{a}_t^{(l)}|\boldsymbol{s}_t^{(l)})} - \frac{\pi_\theta(\check{\boldsymbol{a}}_t|\boldsymbol{s}_t^{(l)})}{\overline{\pi}_\varphi(\check{\boldsymbol{a}}_t|\boldsymbol{s}_t^{(l)})}\right)$$

$$\delta(\theta) = \beta\left(\sum_{t=1}^{T_w}\log\frac{\pi_\theta(\boldsymbol{a}_t^{(w)}|\boldsymbol{s}_t^{(w)})}{\pi_{\text{ref}}(\boldsymbol{a}_t^{(w)}|\boldsymbol{s}_t^{(w)})} - \sum_{t=1}^{T_l}\log\frac{\pi_\theta(\boldsymbol{a}_t^{(l)}|\boldsymbol{s}_t^{(l)})}{\pi_{\text{ref}}(\boldsymbol{a}_t^{(l)}|\boldsymbol{s}_t^{(l)})}\right)$$
$$(15)$$

*then the step-synchronous IT-PO is proposed by*

$$\mathcal{L}_{\text{ss-IT-PO}}(\varphi; \theta) = \mathbb{E}_{(\boldsymbol{x}, \boldsymbol{y}^{(w)}, \boldsymbol{y}^{(l)}) \sim D, \hat{\boldsymbol{A}}, \check{\boldsymbol{A}} \sim \overline{\pi}_\varphi}$$
$$-[(1-\epsilon)\log\sigma(U_{\text{ss}}(\boldsymbol{x}, \boldsymbol{y}^{(w)}, \boldsymbol{y}^{(l)})) + \epsilon\log\sigma(U_{\text{ss}}(\boldsymbol{x}, \boldsymbol{y}^{(l)}, \boldsymbol{y}^{(w)}))]$$
$$(16)$$

Theorem 4.2 establishes that **optimal approximation between** $\overline{\pi}$ **and** $\overline{\pi}_\varphi$ **must preserve their stochastic search strategies' conformity to the BT preference model**. The minimization of Eq.13 is also derived from cDPO that incorporates the margin to tolerate the preference noises possibly introduced by $\hat{\boldsymbol{A}}, \check{\boldsymbol{A}}$ drawn from $\overline{\pi}_\varphi$. It benefits $\overline{\pi}_\varphi$ to generate the high-quality responses that exceed the old data.

**Step-Asynchronous ITS.** MCTS algorithms significantly improve the LM's reasoning ability where the action space of each node focuses on sentence. It holds the asynchronous step between $\overline{\pi}$ and $\pi_\theta$, *i.e.*, $\mathcal{A}_{\text{MCTS}} = \bigcup_{n=1}^N \mathcal{A}^n$, where $\mathcal{A}^n$ is the sentence-level action space with $n$-length tokens and $N$ indicates the maximum length of sentences. Observe that the token-level MDP could be also identified into the sentence level, where $\overline{\pi}, \pi_\theta, \pi_{\text{ref}}$ take their sentence-level actions by generating token sequences ended by "$[\backslash n]$" token (Definition B.3 in Appendix). We re-frame their sentence-level policy as $\overline{\pi}_\varphi^S, \pi_\theta^S, \pi_{\text{ref}}^S$ so that given the state $\boldsymbol{s}_t \in \mathcal{S}_S$ and the sentence-level action $A_t = (a_t^{(i)})_{i=1}^{N_t} \in \mathcal{A}_S$ ($N_t$ indicates the token number in the $t$-th sentence action and $a_t^{(i)}$ indicates the $i$-th token in the $t$-th sentence), and the sentence-level reward $r_S(s_t, A_t) = \sum_{i=1}^{N_t} r(s_t, a_t^{(i)})$.

Given this, we derive the sentence-level ITS optimized by step-asynchronous variant of Eq.4.2:

**Theorem 4.3.** *(Step-Asynchronous-IT-PO) A prompt $x$ drawn from $D$ has a response pairs $y^{(w)} \succ y^{(l)}$ composed of $T_w^S, T_l^S$ sentences, respectively. $A_t^{(w)}, A_t^{(l)}$ denotes the $t$-th sentence in $y^{(w)}, y^{(l)}$, respectively. Suppose the state $s_t^{(w)}, s_t^{(l)}$ transmits along sentence-level MDP (Definition.B.3), $a_t^{(w,i)}/a_t^{(l,i)}$ indicates the $i$-th token in the $A_t^{(w)}/A_t^{(l)}$, $s_t^{(w,i)}/s_t^{(l,i)}$ denotes the sequential context ahead of $a_t^{(w,i)}/a_t^{(l,i)}$ in $y^{(w)}/y^{(l)}$. Suppose that the sentence-level policies $\pi_{\text{ref}}^S, \pi_\theta^S, \overline{\pi}_\varphi^S$ are identified by LM-based token-selection policies $\pi_{\text{ref}}, \pi_\theta, \overline{\pi}_\varphi$, respectively; $\hat{A}_t = (\hat{a}_t^{(1)}, \cdots, \hat{a}_t^{(\hat{N}_t)}) \sim \overline{\pi}_\varphi^S(\cdot|s_t^{(w)}), \hat{s}_t^{(0)} = s_t^{(w)}, \hat{s}_t^{(i+1)} = \mathcal{P}(\hat{s}_t^{(i)}, \hat{a}_t^{(i)}); \check{A}_t = (\check{a}_t^{(1)}, \cdots, \check{a}_t^{(\check{N}_t)}) \sim \overline{\pi}_\varphi^S(\cdot|s_t^{(l)}), \check{s}_t^{(0)} = s_t^{(l)}, \check{s}_t^{(i+1)} = \mathcal{P}(\check{s}_t^{(i)}, \check{a}_t^{(i)})$. It holds*

$$U_{\text{sa}}(x, y^{(w)}, y^{(l)}) = R(x, y^{(w)}) - R(x, y^{(l)})$$
$$= \mu_w^S(\varphi, \theta) - \mu_l^S(\varphi, \theta) + \delta^S(\theta),$$

$$\text{s.t. } \mu_w^S(\varphi, \theta) = -\sum_{t=1}^{T_w^S} \lambda_N(s_t^{(w)}) \left( \prod_{i=1}^{|A_t^{(w)}|} \frac{\pi_\theta(a_t^{(w,i)}|s_t^{(w,i)})}{\overline{\pi}_\varphi(a_t^{(w,i)}|s_t^{(w,i)})} - \prod_{i=1}^{\hat{N}_t} \frac{\pi_\theta(\hat{a}_t^{(i)}|\hat{s}_t^{(i)})}{\overline{\pi}_\varphi(\hat{a}_t^{(i)}|\hat{s}_t^{(i)})} \right)$$

$$\mu_l^S(\varphi, \theta) = -\sum_{t=1}^{T_l^S} \lambda_N(s_t^{(l)}) \left( \prod_{i=1}^{|A_t^{(l)}|} \frac{\pi_\theta(a_t^{(l,i)}|s_t^{(l,i)})}{\overline{\pi}_\varphi(a_t^{(l,i)}|s_t^{(l,i)})} - \prod_{i=1}^{\check{N}_t} \frac{\pi_\theta(\check{a}_t^{(i)}|\check{s}_t^{(i)})}{\overline{\pi}_\varphi(\check{a}_t^{(i)}|\check{s}_t^{(i)})} \right)$$

$$\delta^S(\theta) = \beta \left( \sum_{t=1}^{T_w^S} \sum_{i=1}^{\hat{N}_t} \log \frac{\pi_\theta(\hat{a}_t^{(i)}|\hat{s}_t^{(i)})}{\pi_{\text{ref}}(\hat{a}_t^{(i)}|\hat{s}_t^{(i)})} - \sum_{t=1}^{T_l^S} \sum_{i=1}^{\check{N}_t} \log \frac{\pi_\theta(\check{a}_t^{(i)}|\check{s}_t^{(i)})}{\pi_{\text{ref}}(\check{a}_t^{(i)}|\check{s}_t^{(i)})} \right) \quad (17)$$

*then the step-synchronous IT-PO is proposed by*

$$\mathcal{L}_{\text{sa-IT-PO}}(\varphi; \theta) = \mathbb{E}_{(x, y^{(w)}, y^{(l)}) \sim D, \{\hat{A}_t\}_{t=1}^{T_l^S}, \{\check{A}_t\}_{t=1}^{T_w^S} \sim \overline{\pi}_\varphi}$$
$$-[(1-\epsilon) \log \sigma(U_{\text{ss}}(x, y^{(w)}, y^{(l)})) + \epsilon \log \sigma(U_{\text{ss}}(x, y^{(l)}, y^{(w)}))] \quad (18)$$

## 4.3. Self-Improved Training and Decoding

In Sec.4.1, 4.2, LM $\varphi$ serves as a universal approximator of the ITS policy $\overline{\pi}$ across the state space. Since $\overline{\pi}$ behaves as the stochastic variant of AlphaZero-like tree search, we turn to discuss their common purpose, *i.e.*, distilling $\overline{\pi}_\varphi$ back into the policy $\pi_\theta$ to enhance both their performances.

**Preference Policy Augmentation**. Regular MCTS algorithms used to sample decision-making trajectory to refine the original policy $\pi_\theta$, which is recently treated as a type of generalized policy improvement. Instead, (Grill et al., 2020b) used to distill their ITS policy by $-D_{KL}(\overline{\pi}, \pi_\theta)$. It eases the sampling process but also sacrifices the benefits derived from the high-quality response trajectories.

With this regards, we propose *preference policy augmentation* by resampling $M$ responses $\{y_k\}_{k=1}^K$ generated from each prompt $x \sim \mathcal{D}$ by $\overline{\pi}_\varphi$. $\{y_k\}_{k=1}^K$ are then constructed into pairs with their preference relations evaluated by $U_{\text{ss}}$ or $U_{\text{as}}$, thus, $(y_i \succ y_j)$ holds if either $U_{\text{ss}}(x, y_i, y_j) > 0$ at the token level or $U_{\text{as}}(x, y_i, y_j) > 0$ at the sentence level. After ranking their value by $U_{\text{ss}}$ or $U_{\text{as}}$, the top-$M$ preference pairs are selected for each prompt $x$ in $\mathcal{D}$. Then we collect them across all prompts in $\mathcal{D}$ to construct the ITS

improved dataset $\mathcal{D}^+$. They join with $\mathcal{D}$ to refine the token-selection policy via learning $\theta$ with cDPO. We elaborate the IT-PO algorithm pipelines of its step-synchronous and step-asynchronous cases in our implementation in Appendix.C.

**Self-Improvement in Gradient Analysis**. Through analyzing the TI-PO loss function's gradients with respect to $\varphi$, we demonstrate how $\overline{\pi}_\varphi$ facilitates **the generated responses both diverse and better aligned** with ground-truth preferences. For clarity, our analysis is derived from the step-synchronous ITS using $U_{\text{ss}}$ for preference alignment and focuses on the positive preference logit in $-\log[\sigma(U_{\text{ss}})]$:

$$\nabla_\varphi \left( -\log[\sigma(U_{\text{ss}}(x, y^{(w)})), y^{(l)}] \right)$$
$$= (\nabla_\varphi \mu_w - \nabla_\varphi \mu_l) \cdot \underbrace{\nabla_{\triangle R}(-\log \sigma(\triangle R))}_{\text{higher when reward estimate is wrong by } \varphi}$$

$$\text{s.t. } \nabla_\varphi \mu_w = \sum_{t=1}^{T_w} \lambda_N(s_t^{(w)}) \left( \frac{\pi_\theta(a_t^{(w)}|s_t^{(w)}) \nabla_\varphi \log[\overline{\pi}_\varphi(a_t^{(w)}|s_t^{(w)})]}{\overline{\pi}_\varphi(a_t^{(w)}|s_t^{(w)})} \right.$$
$$\left. - \frac{\pi_\theta(\hat{a}_t|s_t^{(w)}) \nabla_\varphi \log[\overline{\pi}_\varphi(\hat{a}_t|s_t^{(w)})]}{\overline{\pi}_\varphi(\hat{a}_t|s_t^{(w)})} \right)$$

$$-\nabla_\varphi \mu_l = \sum_{t=1}^{T_l} \lambda_N(s_t^{(l)}) \left( -\frac{\pi_\theta(a_t^{(l)}|s_t^{(l)}) \nabla_\varphi \log[\overline{\pi}_\varphi(a_t^{(l)}|s_t^{(l)})]}{\overline{\pi}_\varphi(a_t^{(l)}|s_t^{(l)})} \right.$$
$$\left. + \frac{\pi_\theta(\check{a}_t|s_t^{(l)}) \nabla_\varphi \log[\overline{\pi}_\varphi(\check{a}_t|s_t^{(l)})]}{\overline{\pi}_\varphi(\check{a}_t|s_t^{(l)})} \right). \quad (19)$$

Compared with DPO, the second multiplier observed from the decomposed formula achieves the identical gradient effects. $\nabla_\varphi \log[\overline{\pi}_\varphi(a_t^{(w)}|s_t^{(w)})]$ and $-\nabla_\varphi \log[\overline{\pi}_\varphi(a_t^{(l)}|s_t^{(l)})]$ similarly exist in the first term yet they take the likelihood increase and decrease effects with coefficients $\frac{\pi_\theta(a_t^{(w)}|s_t^{(w)})}{\overline{\pi}_\varphi(a_t^{(w)}|s_t^{(w)})}$ and $\frac{\pi_\theta(a_t^{(l)}|s_t^{(l)})}{\overline{\pi}_\varphi(a_t^{(l)}|s_t^{(l)})}$. It implies that when $\overline{\pi}_\varphi < \pi_\theta$ to the ground-truth preference pairs, IT-PO takes the *aggressive* likelihood influence to align $\overline{\pi}_\varphi$ and $\pi_\theta$ while the *conservative* likelihood influence if $\overline{\pi}_\varphi > \pi_\theta$. More importantly, the implicit tree sampled from the ground-truth preference nodes, *i.e.*, $\hat{a}_t \sim \overline{\pi}_\varphi(\cdot|s_t^{(w)}), \check{a}_t \sim \overline{\pi}_\varphi(\cdot|s_t^{(l)})$ also influence the likelihood: $-\nabla_\varphi \log[\overline{\pi}_\varphi(\hat{a}_t|s_t^{(w)})]$ and $\nabla_\varphi \log[\overline{\pi}_\varphi(\check{a}_t|s_t^{(l)})]$ demonstrates that on account of the ground-truth preference nodes, $\overline{\pi}_\varphi$ learns to search lower preference child nodes from $s_t^{(w)}$ yet higher preference child nodes from $s_t^{(l)}$ with the same conservative-aggressive strategy. Hence $\overline{\pi}_\varphi$ generates certain responses from the preferred context yet explore more diverse responses when the context is not preferred.

**ITS-guided Decoding**. Beyond self-enhanced training, recent study on LLM-based MCTS (Feng et al., 2023) demonstrates promising results in improving the decoding process by MCTS. Motivated from this, we propose the stochastic-tree variant decoding from the spirits of their MCTS-$\alpha$ and MCTS-rollout strategies.

1). **ITS-$\alpha$**. For each initial state $x_{\text{root}}$, the original MCTS-$\alpha$

decoding strategy applied Alpha-like tree search for policy evaluation then backup the visit count $n(\boldsymbol{s}, \boldsymbol{a})$ of the exponential visit distribution $\frac{n(\boldsymbol{s}_t, \boldsymbol{a})^{1/\gamma}}{\sum_{\boldsymbol{a}'} n(\boldsymbol{s}_t, \boldsymbol{a}')^{1/\gamma}}$ to guide decoding. Since $\overline{\pi}_\varphi$ is the universal approximator of $\overline{\pi}$, the stochastic variant of Alpha-like tree search. It is straightfoward to use the exponential version of $\overline{\pi}_\varphi$, $\frac{\exp(\log[\overline{\pi}_\varphi(\boldsymbol{a}|\boldsymbol{s}_t)]/\gamma)}{\sum_{\boldsymbol{a}'} \exp(\log[\overline{\pi}_\varphi(\boldsymbol{a}'|\boldsymbol{s}_t)]/\gamma)}$ (step-synchronous IT-PO) or $\frac{\exp(\log[\overline{\pi}_\varphi^S(\boldsymbol{a}|\boldsymbol{s}_t)]/\gamma)}{\sum_{\boldsymbol{a}'} \exp(\log[\overline{\pi}_\varphi^S(\boldsymbol{a}'|\boldsymbol{s}_t)]/\gamma)}$ (step-asynchronous IT-PO) in our decoding strategy.

2). **ITS-rollout**. When the root $\boldsymbol{x}_{\text{root}}$ and the intermediate state nodes sufficiently differ from the states visited in training, the simulation rounds $N$ behaves more closely as zero to fail the approximation between $\hat{\pi}$ and $\overline{\pi}$ in Theorem.3.3. In this case, $\overline{\pi}_\varphi$ need to be updated to adapt the state $\boldsymbol{x}_{\text{root}}$ as the backup process in MCTS. Due to no value function available, we employ $\pi_\theta$ to implicitly evaluate arbitrary pairs of responses $\boldsymbol{y}_1, \boldsymbol{y}_2$ generated by $\overline{\pi}_\varphi(\cdot|\boldsymbol{x})$, then the sampled responses with preferences evaluated by $\theta$ would join the meta-update of $\overline{\pi}_\varphi$ to the state $\boldsymbol{x}$

$$\varphi' \leftarrow \varphi - \nabla_\varphi^{(\boldsymbol{x}, \boldsymbol{y}_1 \succ \boldsymbol{y}_2)} \mathcal{L}_{\text{sa-IT-PO}}$$
$$\text{s.t. } \boldsymbol{y}_1 \succ \boldsymbol{y}_2 \text{ if } u_\theta(\boldsymbol{x}, \boldsymbol{y}_1, \boldsymbol{y}_2) > 0, \tag{20}$$

which refreshes $\frac{\exp(\log[\overline{\pi}_{\varphi'}(\boldsymbol{a}|\boldsymbol{s}_t)]/\gamma)}{\sum_{\boldsymbol{a}'} \exp(\log[\overline{\pi}_{\varphi'}(\boldsymbol{a}'|\boldsymbol{s}_t)]/\gamma)}$ to facilitate our decoding strategy. ITS-rollout can be treated as test-time-training version of ITS-$\alpha$. Its implementation first updates $\varphi'$ by (20), then use ITS-$\alpha$ with $\varphi'$ to facilitate the decoding process. In our experiment, 8 responses for each test prompt were generated to achieve self-training, resulting the extra 0.5~1 hour as the decoding warm-up stage.

Provided the decoding probabilities obtained by ITS-rollout, the BF-S strategy (Breath-first Search) can be applied based on the *implicit advantage function* solely approximated by $\overline{\pi}_\varphi$ and $\pi_\theta$. More specifically,

$$
\begin{aligned}
A^{\pi_\theta}(\boldsymbol{a}, \boldsymbol{s}) =& Q^{\pi_\theta}(\boldsymbol{a}, \boldsymbol{s}) - \mathbb{E}[Q^{\pi_\theta}(\boldsymbol{a}', \boldsymbol{s})|\boldsymbol{a}' \sim \pi_\theta(\cdot|\boldsymbol{s})] \\
\approx& Q^{\pi_\theta}(\boldsymbol{a}, \boldsymbol{s}) - \frac{1}{N} \sum_{\boldsymbol{a}' \sim \pi_\theta(\cdot|\boldsymbol{s})} Q^{\pi_\theta}(\boldsymbol{a}', \boldsymbol{s}) \\
=& \Big(\alpha(\boldsymbol{s}) - \lambda_N(\boldsymbol{s}) \frac{\pi_\theta(\boldsymbol{a}|\boldsymbol{s})}{\overline{\pi}(\boldsymbol{a}|\boldsymbol{s})}\Big) \\
& - \Big(\frac{1}{N} \sum_{\boldsymbol{a}' \sim \pi_\theta(\cdot|\boldsymbol{s})} \big(\alpha(\boldsymbol{s}) - \lambda_N(\boldsymbol{s}) \frac{\pi_\theta(\boldsymbol{a}'|\boldsymbol{s})}{\overline{\pi}(\boldsymbol{a}'|\boldsymbol{s})}\big)\Big) \\
=& -\lambda_N(\boldsymbol{s}) \Big(\frac{\pi_\theta(\boldsymbol{a}|\boldsymbol{s})}{\overline{\pi}(\boldsymbol{a}|\boldsymbol{s})} - \frac{1}{N} \sum_{\boldsymbol{a}'} \frac{\pi_\theta(\boldsymbol{a}'|\boldsymbol{s})}{\overline{\pi}(\boldsymbol{a}'|\boldsymbol{s})}\Big) \\
\approx& -\lambda_N(\boldsymbol{s}) \Big(\frac{\pi_\theta(\boldsymbol{a}|\boldsymbol{s})}{\overline{\pi}_\varphi(\boldsymbol{a}|\boldsymbol{s})} - \frac{1}{N} \sum_{\boldsymbol{a}'} \frac{\pi_\theta(\boldsymbol{a}'|\boldsymbol{s})}{\overline{\pi}_\varphi(\boldsymbol{a}'|\boldsymbol{s})}\Big),
\end{aligned}
\tag{21}
$$

where $N$ indicates how many responses drawn from $\pi_\theta$ to approximate the advantage function. Such decoding strategy is closely related with wisdom of group-relative policy optimization (GRPO), a well-known value-free RL algorithm employed to train Deepseek (Shao et al., 2024).

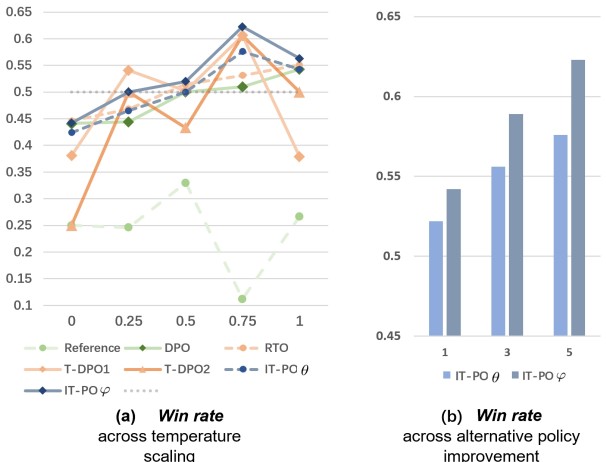

*Figure 2.* **Win rate** measured by GPT-4 via the consistent prompts in previous studies: (a) **Win rate** of baselines decoding with different temperatures; (b) **Win rate** of $\pi_\theta$ and $\overline{\pi}_\varphi$ across the alternative phases of their preference policy distillation.

## 5. Experiments

In this section, we demonstrate the superiority of IT-PO from the step-synchronous (Theorem.4.2) and step-asynchronous (Theorem.4.3) perspectives. In the step-synchronous cases, IT-PO trains $\overline{\pi}_\varphi$ to perform token-level ITS in order to provide the fine-grained human preference alignment; in the step-asynchronous scenarios, we evaluate $\overline{\pi}_\varphi$ trained by IT-PO to perform sentence-level search in step-asynchronous scenarios, *i.e.*, mathematical reasoning and planning tasks where LLM-based MCTS algorithms are extensively used.

### 5.1. Experiments on Token-level Preference Alignment

Anthropic HH dataset (Bai et al., 2022)[1] consists of 170k dialogues between a human and an automated assistant, each of which presents as a history with alternative responses with respect to different preferences annotated by humans. We conduct the conventional evaluation setup using Pythia 2.8 (Biderman et al., 2023) as the base model, then each dialogue with its preferred completion in Anthropic HH training set is incorporated for supervised fine-tuning to derive three LMs: the parameter-frozen $\pi_{\text{ref}}$ and parameter-initialized $\pi_\theta$, $\overline{\pi}_\varphi$. In terms of preference alignment on token-level MDP, SS-IT-PO with respect to the LMs $\theta$, $\varphi$ are alternatively trained, then the IT-PO variants both compared with token-level DPO-family baselines, *i.e.*, DPO (Rafailov et al., 2024a), RTO (Zhong et al., 2024a), TDPO$_1$ and TDPO$_2$ (Zeng et al., 2024a).

The experiment primarily aims for three evaluation metrics: 1). **Accuracy**: we adopt the evaluation split in (Zeng et al., 2024a) to train all models then evaluate their performance

---

[1]https://huggingface.co/datasets/Anthropic/hh-rlhf

*Table 1.* Comparison in terms of the trade-off between Alignment (accuracy) and Diversity (entropy) on the Anthropic HH dataset. The ↑ indicates higher values are preferable.

| Method | Alignment | Diversity |
|---|---|---|
| | Acc (%) ↑ | Ent ↑ |
| DPO | 59.43 | 3.196 |
| RTO | 61.43 | 3.314 |
| TDPO$_1$ | 60.08 | 4.727 |
| TDPO$_2$ | **67.33** | **4.915** |
| IT-PO $\theta$ (ours) | 67.75 | 4.564 |
| IT-PO $\varphi$ (ours) | **69.12** | **5.315** |

in terms of the accuracy on the generated responses relative to chosen completions in the test dataset; 2). **Diversity**: Nucleus sampling with $p = 0.95$ to generate 25 responses then the predictive entropy across the responses indicates the generation diversity. 3).**Win rate**: all baseline approaches are evaluated through GPT-4 against the chosen responses in the test set, so that $> 50\%$ implies the human preference alignment achieved in their performances.

Both LMs $\theta$ and $\varphi$ utilize **ITS**-$\alpha$ to enhance the decoding strategy. The exponential rate $\gamma =1$ consistently across the evaluation of three metrics. This choice aligns with MCTS-$\alpha$, which employs token-level tree search in alignment tasks. **Accuracy** and **Diversity** across the baselines are presented in Table 5.1. TDPO$_2$ demonstrates competitive performance, achieving alignment accuracy comparable to IT-PO $\theta$, even exhibiting superior diversity. On the other hand, IT-PO $\varphi$, trained as the stochastic tree policy, outperforms TDPO$_2$ by a significant margin in both alignment and diversity metrics. It is because that LM $\theta$ in our framework performs more likely as a policy evaluator trained to boost the evolution of stochastic tree policy $\overline{\pi}_\varphi$.

To verify our assumption, we further observe their **Win rate** illustrated in Figure.2(a). Through scaling the temperature during inference, T-DPOs are found quite sensitive to the temperature, with lower performance than most baselines when the temperature value are extreme. Instead, both IT-PO $\theta$ and IT-PO $\varphi$ consistently performs with robust value over $50\%$ in 7 out of 10 cases where the other baselines underperform or even fail in the qualified **Win rate** level. In Figure.2 (b), we further observe the **Win rate** progress when IT-PO $\theta$ and IT-PO $\varphi$ by through their alternative policy distillation. When no alternative strategy used (iter = 1), the win rate of IT-PO $\theta$ and IT-PO $\varphi$ are solely on par with the win rates of DPO and RTO, respectively, which largely underperform T-DPO variants. While after five iterations of alternative preference policy distillation, **Win rate**s in IT-PO $\theta$ and IT-PO $\varphi$ are both incredibly improved to exceed their early, even their teachers' performances. These observations are consistent with our analysis to $\varphi$'s gradients.

*Table 2.* Performance comparison of different methods on GSM8k and Game24 datasets

| Setting | Method | Performance(%) / # Tokens | | | |
|---|---|---|---|---|---|
| | | GSM8k | | Game24 | |
| Path@1 | CoT-greedy | 41.4 | 98 | 12.7 | 76 |
| | BFS-V | **52.5** | 485 | 64.8 | 369 |
| | MCTS-$\alpha$ | 51.9 | 561 | 63.3 | 412 |
| | MCTS-Rollout | 47.8 | 3.4k | **71.3** | 670 |
| | ITS-$\alpha$ (**ours**) | **53.2** | 561 | 67.6 | 380 |
| | ITS-Rollout (**ours**) | 51.6 | 3.8k | **73.2** | 646 |
| Equal-Token | CoT-SC$_{MAJ}$ | 46.8 | 500 | 14.6 | 684 |
| | CoT-SC$_{ORM}$ | **52.3** | 500 | 50.6 | 684 |
| | BFS-V$_{ORM}$ | - | - | **70.90** | 1.6k |
| | MCTS$_{ORM}$ | - | - | 69.34 | 649 |
| | ITS-$\alpha$ (**ours**) | - | - | 70.64 | 1.6k |
| | ITS-Rollout (**ours**) | - | - | **71.42** | 698 |

### 5.2. Experiments on Mathematical Tasks

Our second experimental suite includes the tasks of mathematical reasoning on GSM8K (Cobbe et al., 2021), MATH (Hendrycks et al., 2021) and mathematical planning on Game24 (Cobbe et al., 2021). Although the benchmarks are famous in evaluating LLM's reasoning capability by PRM (process reward model) (Lightman et al., 2023) and MCTS-based methods (Feng et al., 2023), they are nontrivial for DPO family since explicit value function can not be skipped to execute MCTS reasoning strategies.

**Baselines and Evaluation**. Since most DPO-based approaches can not adapt to LLM reasoning without explicit value modeling, we are more interested in the comparison between TI-PO and state-of-the-art tree-search (TS) LLM and Chain-of-Thought (CoT) (Wei et al., 2022) baselines in (Feng et al., 2023). More specifically, the evaluated baselines for GSM8K and Game24 are derived from LLAMA-7b [2] base model, specifically include CoT-greedy (greedy value search by CoT), BFV (Breath-first search with their learned value function), MCTS-$\alpha$ (AlphaZero-like tree search with their learned value function), MCTS-rollout (MCTS-$\alpha$ variant that allows the backup process happen in the intermediate step). As a counterpart, we incorporate the policies $\pi_\theta$ and $\overline{\pi}_\varphi$ with their LMs fine-tuned by cDPO (Eq.13) and step-asynchronous IT-PO objective (Eq.17), respectively, then achieve their alternative post-training via preference policy augmentation. After five iterations, the policy models jointly facilitate the decoding processes via **ITS**-$\alpha$ and **ITS-rollout** strategies. All baselines are evaluated in the single path setup via Path@1 and Equal-Token, the latter try to

---

[2] https://huggingface.co/meta-llama/Llama-2-7b

*Table 3.* Path@1 metric on Game24 with different node size.

| Method | Performance(%) / # Tokens | | |
|---|---|---|---|
| | width=6 | width=20 | width=50 |
| MCTS-$\alpha$ | 41.6 243 | 63.3 412 | 74.5 573 |
| MCTS-Rollout | 43.8 401 | 71.3 670 | 80.7 833 |
| BFS-V | 43.2 206 | 64.8 370 | 74.6 528 |
| ITS-$\alpha$ (**ours**) | 46.1 267 | 67.6 380 | 75.0 647 |
| ITS-Rollout (**ours**) | **48.9** 489 | **73.2** 646 | **81.8** 954 |

*Table 4.* The experimental results in MATH. All baselines employed Qwen1.5-32B as their base models.

| Baselines/Decoding | greedy CoT | -$\alpha$ | -rollout |
|---|---|---|---|
| Qwen1.5-32B | 36.1 | - | - |
| MCTS- (Qwen1.5-32B) | - | 36.0 | 36.7 |
| ITS- ($\varphi$=Qwen1.5-32B) | - | 39.8 | 40.2 |
| ITS- ($\varphi$=Qwen1.5-32B) | - | 37.9 | 38.2 |

compare the results with the similar scale computation consumption. For the evaluation on MATH, we construct the training set integrated with the training splits of GSM8K and MATH, then consider Qwen1.5-32B (Team, 2024) as our base model. Beyond this, we also introduced LLAMA3.1-8B as an alternative of $\varphi$ to verify whether $\varphi$ can be replaced by a smaller LLM. To this, we have the evaluated LLMs ($\theta,\varphi$) defined by (Qwen1.5-32B,Qwen1.5-32B) and (Qwen1.5-32B,LLAMA3-8B), respectively. We also employed greedy CoT (3 shots), MCTS-$\alpha$, and MCTS-rollout for their comparison.

**Implementation**. Distinct from human preference data with pairwise responses, reasoning tasks only consist of question prompts and its correct solution responses. It motivates us to reconfigure their data to adapt our preference optimization regime. Beyond this, since our "tree-search" strategy is indeed a stochastic policy, the search depth and breadth for training LLM $\varphi$ were not limited in all datasets, while their decoding procedures *i.e.*, ITS-$\alpha$ and ITS-rollout, inherited the pruning setup derived from the MCTS baselines to address the heavy computation while the tree search runs for reasoning-oriented inference. Details refer to Appendix.C.

**Results**.According to the GSM8k results (Table 2), when examining Path@1 performance across all baseline methods, tree-search algorithms (excluding our proposed methods) don't show significant advantages over standard CoT approaches. MCTS variants actually perform worse than BFV, despite their higher computational requirements. However, the Game24 results tell a different story. In this task, CoT approaches perform poorly, largely because Game24's structure allows for wider and deeper search trees, which better showcases the strengths of tree-search algorithms (Table B.5). But interestingly, **ITS-$\alpha$** and **ITS-rollout** consistently outperform other approaches, regardless of the default tree width and depth limits during inference.

This superior performance can be attributed to our IT-PO training regime. Unlike traditional approaches that slowly build sparse trees based on visit counts, our approximated $\overline{\pi}$ encourages broader exploration across the entire action space. This approach is computationally efficient since IT-PO only needs to sample the next sentence (rather than the

complete reasoning path) from preferred/dispreferred contexts (as defined by $\hat{A}_t$ and $\check{A}_t$ in Eq. 4.3). This enables faster search and backup operations compared to standard MCTS. As a result, even with bounded search width and depth during inference, the stochastic policy demonstrates robust performance due to effective training simulation. In order to support our claim, we further ablate the maximum tree width and node size during inference to observe the performance variation across different baselines. As shown in Table.3,4, expanding the node size and tree width significantly boosts regular TS-LLM performances. While ITS-$\alpha$ and ITS-Rollout have already achieved impressive results with the small node size and and tree width. Beyond this, they also enjoy the performance growth along with the increasing exploration space.

In Table.5, we presented the evaluation in MATH that contains more difficult mathematical reasoning tasks. In terms of the greedy CoT results in LLAMA3-8B (20.5), some observations can be found in the table. First, we note that the MCTS strategy derived from GSM8K is unreliable in MATH, probably due to the conflict between the problem complexity and limited search width and depth during training, while ITS-$\alpha$ and ITS-rollout did not suffer from this problem. Second, with a weaker model (LLAMA3-8B) to implement LLM $\varphi$, IT-PO still enabled the improvement of LLM-based reasoning. It implies the feasibility of IT-PO. Specifically, despite avoiding value modeling, IT-PO introduces the extra LM $\varphi$ instead of the single LM $\theta$ in standard DPO variants, leading to the increased computational and memory requirement. While the evidences in Table.5 demonstrated that using LM $\varphi$ with the significantly smaller and weaker base model than LM $\theta$, ITS-($\theta$ =Qwen1.5-32B, $\varphi$ = LLAMA3-8B) still yields the results very competitive with ITS-($\theta$ = Qwen1.5-32B, $\varphi$ = Qwen1.5-32B).

## 6. Concluding Remark

This paper presents a new RL-free methodology to equip DPO with MCTS interpreted as stochastic policy to better align LMs with human preferences, and simultaneously outperforms MCTS-based LLM reasoning baseline methods in both mathematical reasoning and planning tasks.

## Acknowledgement

The research was supported in part by Guangdong S&T Programme (Grant No. 2024B0101010003); in part by The Major Key Project of PCL (No. PCL2024A04, PCL2025A02); in part by National Natural Science Foundation of China (NSFC) under Grant No.62206110, 62176103, 62377208, and 62276114; in part by the Science and Technology Planning Project of Guangzhou under grants 2024A04J9896, 2025A03J3565.

In particular, we thank all the reviewers for their constructive suggestions that help to improve this work.

## Impact Statement

This work advances preference-based learning for large language models through improved exploration mechanism, which has implications for AI alignment and safer deployment of language technologies. By enhancing LLMs' reasoning and long-form generation capabilities through more principled exploration and preference learning, our approach could lead to more reliable and controllable language models. However, improved reasoning capabilities could also enable more sophisticated text generation that may be misused. The integration of MCTS with preference learning represents a step toward more transparent optimization of language model behavior, though careful consideration should be given to the selection of preference data to avoid encoding harmful biases. We believe the technical advances presented here can contribute positively to the development of more capable and aligned language models when deployed thoughtfully with appropriate safeguards and oversight.

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

# A. Related Work

The integration of search-based optimization techniques in language model alignment has significantly advanced AI alignment, with Reinforcement Learning with Human Feedback (RLHF)(Bai et al., 2022; Wang et al., 2023; Kirk et al., 2023; Dong et al., 2024) being a widely used approach. RLHF employs reward models trained from human feedback to optimize model behavior through reinforcement learning, typically using Proximal Policy Optimization (PPO) (Zhong et al., 2024b). However, RLHF has been criticized for instability, sample inefficiency, and over-optimization issues (Engstrom et al., 2020; Liu et al., 2024b; Chen et al., 2023; 2017; 2024c). To address these challenges, methods like Reward Ranked FineTuning (RAFT) (Dong et al., 2023) and Rank Responses to align Human Feedback (RRHF) (Yuan et al., 2023) have been proposed to refine ranking-based optimization without explicit reinforcement learning. More recently, Direct Preference Optimization (DPO) (Rafailov et al., 2024c; Amini et al., 2024) has emerged as an alternative, allowing language models to be aligned directly from human preference data without requiring a reward model.

Unlike PPO, which operates within a reinforcement learning framework by optimizing policies through reward feedback, DPO reformulates the alignment problem as a supervised learning task, making policy updates more stable while maintaining alignment with human preferences. Further refinements, such as processing the paragraph at token level with methods such as TDPO (Zeng et al., 2024b), T-REG (Zhou et al., 2024), TPPO (Ouyang et al., 2024), and TIS-DPO (Liu et al., 2024a), improve efficiency by incorporating token-wise adjustments, improving preference alignment in a manner that contrasts with RLHF's reliance on policy gradient updates.

Monte Carlo Tree Search (MCTS) has been extensively applied in decision-making tasks, particularly in game-playing AI, as demonstrated by its success in AlphaGo (Silver et al., 2017b), AlphaZero (Silver et al., 2017a), and MuZero (Schrittwieser et al., 2020). Recent advancements have expanded its application to large language models (LLMs) for structured text generation, such as in AlphaZero-like Tree-Search for LLMs (TS-LLM) (Feng et al., 2023). Similarly, Xie et al. (Xie et al., 2024c) propose an iterative preference learning approach, using MCTS to refine step-wise reasoning capabilities in LLMs. Further developments by Wang et al. (Wang et al., 2024b) introduce self-improvement techniques where LLMs leverage MCTS for preference-guided reinforcement learning, while Chen et al. (Chen et al., 2024b) focus on step-level value preference optimization, allowing fine-grained preference learning at intermediate steps. Zhang et al. (Zhang et al., 2024b) extend this concept with chain preference optimization, improving long-range decision-making for complex reasoning tasks. Lastly, Liao et al. (Liao et al., 2024) introduce Tree-based Preference Optimization (TPO), which integrates MCTS with preference alignment techniques to refine LLM outputs progressively. Despite these advancements, a key challenge remains: these approaches still require learning an explicit reward value function, which is critical for optimizing the search process and improving the efficiency of LLM training and inference.

Our work proposes *Implicit Tree Search (ITS)*, which integrates stochastic policy optimization with AlphaZero-like search principles while eliminating explicit tree structures. ITS leverages reversed KL-divergence constraints and a stochastic sampling policy to enhance exploration expressiveness, addressing cold-start issues common in MCTS-based methods. Similar ideas have been explored in Monte Carlo-based regularized policy optimization (Grill et al., 2020a), (Wang et al., 2024a) they use pairwise training framework that enables LLMs to self-improve through MCTS behavior but ITS extends these principles to preference learning in language models.

Furthermore, our approach aligns with broader research on AI safety and preference-based learning (Yuan et al., 2024; Xie et al., 2024a; Chen et al., 2024a; Mitchell, 2023a). ITS represents a novel *search-and-learn* paradigm that improves structured reasoning in language models without relying on explicit tree expansion, bridging the gap between MCTS and modern alignment techniques.

### C.1. Fundamentals of Monte Carlo Tree-Search Methods

Monte Carlo Tree Search (MCTS) has been widely adopted as an effective strategy for solving problems requiring sequential decision-making and planning. Traditional MCTS operations, as introduced by (Kocsis & Szepesvári, 2006) and (Coulom, 2006), include four key steps: selection, expansion, simulation, and backpropagation. However, these operations can be adapted for more advanced frameworks like AlphaZero (Silver et al., 2017b), which incorporates a learned value function and policy network to guide the search process.

To address challenges in balancing exploration and exploitation, we utilize a variant of MCTS with a modified Predictor Upper Confidence Tree (PUCT) algorithm (Rosin, 2011). The algorithm selects actions $a_t$ at each node $s_t$ as follows:

$$a_t = \arg\max_a \big( Q(s_t, a) + U(s_t, a) \big),$$

where $U(s_t, a)$ is calculated using the formula:

$$U(s, a) = c_{\text{puct}} \cdot \pi_\theta(s, a) \cdot \frac{\sqrt{\sum_b N(s, b)}}{1 + N(s, a)}.$$

Here, $N(s, a)$ represents the visit count of action $a$ at node $s$, and $c_{\text{puct}}$ is a constant controlling exploration, as defined by:

$$c_{\text{puct}} = \log\left(\frac{\sum_b N(s, b) + c_{\text{base}} + 1}{c_{\text{base}}}\right) + c_{\text{init}}.$$

**Node Expansion and Assessment:** Upon reaching a leaf node $s_L$, if it is not terminal, the tree is expanded by generating possible successor nodes. The value of the leaf node is then estimated using a neural network. For terminal nodes, a reward function $R(s_L)$ is used, or an Outcome Reward Model (ORM) serves as an approximation (Uesato et al., 2022).

**Value Propagation:** Once a leaf node is assessed, the computed values are propagated back through the path $s_0, s_1, \ldots, s_L$. For each node, the visit count is updated as:

$$N(s_t, a) = N(s_t, a) + 1,$$

and the cumulative action value is updated as:

$$W(s_t, a) = W(s_t, a) + v(s_L).$$

The mean action value is then computed as:

$$Q(s_t, a) = \frac{W(s_t, a)}{N(s_t, a)}.$$

This combination of learned value functions and search-based methods enables efficient exploration of large decision spaces, as demonstrated in AlphaZero (Silver et al., 2017b) and its applications to tree-search-guided language models (Yao et al., 2023; Hao et al., 2023).

## B. Proofs.

### B.1. Proof of Proposition 3.1

*Proof.* Given $r \in \mathbb{R}$ and $\forall \boldsymbol{a}' \in \mathcal{A}_{\text{MCTS}}/\{\boldsymbol{a}\}$, $n(\boldsymbol{s}, \boldsymbol{a}')$ are fixed, the probability of empirical visit distribution to the action $\boldsymbol{a}$ denotes as $\frac{1+n(\boldsymbol{s},\boldsymbol{a})^r}{|\mathcal{A}_{\text{MCTS}}|+\sum_{a' \in \mathcal{A}_{\text{MCTS}}} n(\boldsymbol{s},\boldsymbol{a}')^r}$. It holds

$$\frac{1 + n(\boldsymbol{s}, \boldsymbol{a})^r}{|\mathcal{A}_{\text{MCTS}}| + \sum_{a' \in \mathcal{A}_{\text{MCTS}}} n(\boldsymbol{s}, \boldsymbol{a}')^r} = 1 - \frac{1 - (|\mathcal{A}_{\text{MCTS}}| + \sum_{a' \in \mathcal{A}_{\text{MCTS}}/\{a\}} n(\boldsymbol{s}, \boldsymbol{a}')^r)}{|\mathcal{A}_{\text{MCTS}}| + \sum_{a' \in \mathcal{A}_{\text{MCTS}}/\{a\}} n(\boldsymbol{s}, \boldsymbol{a}')^r + n(\boldsymbol{s}, \boldsymbol{a})^r} \tag{22}$$

Since $|\mathcal{A}_{\text{MCTS}}| + \sum_{a' \in \mathcal{A}_{\text{MCTS}}/\{a\}} n(\boldsymbol{s}, \boldsymbol{a}')^r$ is constant, the exponential empirical visit distribution only changes along with $n(\boldsymbol{s}, \boldsymbol{a}')^r$ changes. Due to $|\mathcal{A}_{\text{MCTS}}| > 1$ as "tree" definition, $|\mathcal{A}_{\text{MCTS}}| + \sum_{a' \in \mathcal{A}_{\text{MCTS}}/\{a\}} n(\boldsymbol{s}, \boldsymbol{a}')^r > 1$. Note that $f(x) = 1 - \frac{1-c}{c+x}$ is bijective with respect to $c > 1$ and when $r \neq 0$, $\{n^r : n \in \mathbb{Z} \cup \{0\}\}$ holds the consistent cardinality with $\mathbb{Z}$. The proposition has been proved. $\qquad\square$

### B.2. Proof of Proposition 3.2

*Proof.* To prove the proposition, we only need to prove the following lemma:

**Lemma B.1.** *Given positive integers $a, b$, and $\{c_i\}_{i=1}^a$, define*

$$f(r) = \frac{1 + b^r}{a + 1 + \sum_{i=1}^a c_i^r + b^r}.$$

*We claim that for all real $r$, the value of $f(r)$ lies strictly within the open interval $\left(\frac{1}{a+1}, 1\right)$*

*Proof.* 1. **As** $r \to -\infty$**:** Since $b^r \to 0$ and $c_i^r \to 0$ for each $i$, we have

$$\lim_{r \to -\infty} f(r) = \lim_{r \to -\infty} \frac{1 + b^r}{a + 1 + \sum_{i=1}^a c_i^r + b^r} = \frac{1 + 0}{a + 1 + 0 + 0} = \frac{1}{a + 1}.$$

Thus $f(r)$ never goes below $1/(a+1)$.

2. **As** $r \to +\infty$**:** Let $M = \max\{b, c_1, c_2, \ldots, c_a\}$. For sufficiently large $r$, $M^r$ dominates $b^r$ and each $c_i^r$, so the numerator and denominator in $f(r)$ are asymptotically proportional to $M^r$, giving

$$\lim_{r \to +\infty} f(r) = 1.$$

Since $b^r$ and $c_i^r$ increase with $r$, one can show $f(r)$ itself is strictly increasing in $r$. Consequently, its image over $r \in \mathbb{R}$ is precisely

$$\left( \tfrac{1}{a+1}, 1 \right).$$

**Counterexample.** Because $f(r)$ is bounded below by $\frac{1}{a+1}$, any real $x$ with

$$0 < x < \frac{1}{a + 1}$$

cannot be realized by $f(r)$. For instance, choose

$$x_0 = \frac{1}{2(a + 1)}.$$

We observe

$$0 < \frac{1}{2(a + 1)} < \frac{1}{a + 1},$$

but there is no real $r$ for which $f(r) = x_0$. Hence, values in $\left( 0, \frac{1}{a+1} \right)$ are not attainable. □

Set $a = |\mathcal{A}_{\mathsf{MCTS}}| - 1$, $b = n(s, a)$, and $c_i = n(s, a_i), \forall a_i \in \mathcal{A}_{\mathsf{MCTS}}/\{a\}$, the proposition has been proved. □

### B.3. Proof of Lemma 4.1

*Proof.* As discussed in Lemma.3.4,

$$\overline{\pi}_\varphi(\boldsymbol{a}_t | \boldsymbol{s}_t) = \lambda_N(\boldsymbol{s}_t) \frac{\pi_\theta(\boldsymbol{a}_t | \boldsymbol{s}_t)}{\alpha(\boldsymbol{s}_t) - Q^{\pi_\theta}(\boldsymbol{s}_t, \boldsymbol{a}_t)} \leftrightarrows r(\boldsymbol{s}_t, \boldsymbol{a}_t) = \alpha(\boldsymbol{s}_t) - V^{\pi_\theta}(\boldsymbol{s}_t) - \lambda_N(\boldsymbol{s}_t) \frac{\pi_\theta(\boldsymbol{a}_t | \boldsymbol{s}_t)}{\overline{\pi}_\varphi(\boldsymbol{a}_t | \boldsymbol{s}_t)}, \tag{23}$$

where $Q^{\pi_\theta}(\boldsymbol{s}_t, \boldsymbol{a}_t) = r(\boldsymbol{s}_t, \boldsymbol{a}_t) + V^{\pi_\theta}(\boldsymbol{s}_t)$. Provided $(\boldsymbol{x}, \boldsymbol{y}^{(w)})$ denoting a pair of prompt and its preferred response, we construct the preferred state-action trajectory along with the token-level MDP, *i.e.*, $\{(\boldsymbol{s}_t^{(w)}, \boldsymbol{a}_t^{(w)})\}_{t=1}^{T_w}$ *w.r.t.* $\boldsymbol{s}_t^{(w)} = (\boldsymbol{x}, \boldsymbol{y}_{<t}^{(w)})$ and $\boldsymbol{a}_t^{(w)} = y_t^{(w)}$. Then based on the definition, for the $t$-th state in the preferred state-action trajectory, we sample another action $\hat{a}_i$ from the parameterized ITS exploration policy $\overline{\pi}_\varphi$ *i.e.*, $\hat{a}_t \sim \overline{\pi}_\varphi(\cdot | \boldsymbol{s}_t^{(w)})$. For the preferred state-action trajectory, Eq.23 holds at each step so that

$$R(\boldsymbol{x}, \boldsymbol{y}^{(w)}) = \sum_{t=1}^{T_w} r(\boldsymbol{s}_t^{(w)}, \boldsymbol{a}_t^{(w)}) = \sum_{t=1}^{T_w} \left( \alpha(\boldsymbol{s}_t^{(w)}) - V^{\pi_\theta}(\boldsymbol{s}_t^{(w)}) - \lambda_N(\boldsymbol{s}_t^{(w)}) \frac{\pi_\theta(\boldsymbol{a}_t^{(w)} | \boldsymbol{s}_t^{(w)})}{\overline{\pi}_{\varphi^*}(\boldsymbol{a}_t^{(w)} | \boldsymbol{s}_t^{(w)})} \right) \tag{24}$$

and $\forall t \in T_w$,

$$r(\boldsymbol{s}_t^{(w)}, \hat{\boldsymbol{a}}_t) = \alpha(\boldsymbol{s}_t^{(w)}) - V^{\pi_\theta}(\boldsymbol{s}_t^{(w)}) - \lambda_N(\boldsymbol{s}_t^{(w)}) \frac{\pi_\theta(\hat{\boldsymbol{a}}_t | \boldsymbol{s}_t^{(w)})}{\overline{\pi}_{\varphi^*}(\hat{\boldsymbol{a}}_t | \boldsymbol{s}_t^{(w)})}. \tag{25}$$

Therefore

$$
\begin{aligned}
R(\boldsymbol{x}, \boldsymbol{y}^{(w)}) - \sum_{t=1}^{T_w} r(\boldsymbol{s}_t^{(w)}, \hat{\boldsymbol{a}}_t) &= \sum_{t=1}^{T_w} \left( \alpha(\boldsymbol{s}_t^{(w)}) - V^{\pi_\theta}(\boldsymbol{s}_t^{(w)}) - \lambda_N(\boldsymbol{s}_t^{(w)}) \frac{\pi_\theta(\boldsymbol{a}_t^{(w)}|\boldsymbol{s}_t^{(w)})}{\overline{\pi}_{\varphi^*}(\boldsymbol{a}_t^{(w)}|\boldsymbol{s}_t^{(w)})} \right) \\
&\quad - \sum_{t=1}^{T_w} \left( \alpha(\boldsymbol{s}_t^{(w)}) - V^{\pi_\theta}(\boldsymbol{s}_t^{(w)}) - \lambda_N(\boldsymbol{s}_t^{(w)}) \frac{\pi_\theta(\hat{\boldsymbol{a}}_t|\boldsymbol{s}_t^{(w)})}{\overline{\pi}_{\varphi^*}(\hat{\boldsymbol{a}}_t|\boldsymbol{s}_t^{(w)})} \right) \\
&= -\lambda_N(\boldsymbol{s}_t^{(w)}) \left( \frac{\pi_\theta(\boldsymbol{a}_t^{(w)}|\boldsymbol{s}_t^{(w)})}{\overline{\pi}_{\varphi^*}(\boldsymbol{a}_t^{(w)}|\boldsymbol{s}_t^{(w)})} - \frac{\pi_\theta(\hat{\boldsymbol{a}}_t|\boldsymbol{s}_t^{(w)})}{\overline{\pi}_{\varphi^*}(\hat{\boldsymbol{a}}_t|\boldsymbol{s}_t^{(w)})} \right).
\end{aligned}
\tag{26}
$$

Therefore $R(\boldsymbol{x}, \boldsymbol{y}^{(w)}) = \sum_{t=1}^{T_w} \left( r(\boldsymbol{s}_t^{(w)}, \hat{\boldsymbol{a}}_t) - \lambda_N(\boldsymbol{s}_t^{(w)}) \left( \frac{\pi_\theta(\boldsymbol{a}_t^{(w)}|\boldsymbol{s}_t^{(w)})}{\overline{\pi}_\varphi(\boldsymbol{a}_t^{(w)}|\boldsymbol{s}_t^{(w)})} - \frac{\pi_\theta(\hat{\boldsymbol{a}}_t|\boldsymbol{s}_t^{(w)})}{\overline{\pi}_\varphi(\hat{\boldsymbol{a}}_t|\boldsymbol{s}_t^{(w)})} \right) \right).$

The dispreferred state-action trajectory from $(\boldsymbol{x}, \boldsymbol{y}^l)$ can be also constructed in the same routine, *i.e.*, $\{(\boldsymbol{s}_t^{(l)}, \boldsymbol{a}_t^{(l)})\}_{t=1}^{T_l}$ *w.r.t.* $\boldsymbol{s}_t^{(l)} = (\boldsymbol{x}, \boldsymbol{y}_{<t}^{(l)})$ and $\boldsymbol{a}_t^{(l)} = y_t^{(l)}$. For the $i$-th state in the dispreferred state-action trajectory, we sample another action $\check{a}_i$ from $\overline{\pi}_\varphi$ *i.e.*, $\check{a}_i \sim \overline{\pi}_\varphi(\cdot|\boldsymbol{s}_t^{(l)})$. For the dispreferred state-action trajectory, it holds the mirror formulations of Eq.24-26 , then from the same deduction, $R(\boldsymbol{x}, \boldsymbol{y}^{(l)}) = \sum_{t=1}^{T_l} \left( r(\boldsymbol{s}_t^{(l)}, \check{\boldsymbol{a}}_t) - \lambda_N(\boldsymbol{s}_t^{(l)}) \left( \frac{\pi_\theta(\boldsymbol{a}_t^{(l)}|\boldsymbol{s}_t^{(l)})}{\overline{\pi}_\varphi(\boldsymbol{a}_t^{(l)}|\boldsymbol{s}_t^{(l)})} - \frac{\pi_\theta(\check{\boldsymbol{a}}_t|\boldsymbol{s}_t^{(l)})}{\overline{\pi}_\varphi(\check{\boldsymbol{a}}_t|\boldsymbol{s}_t^{(l)})} \right) \right)$ is hold.  □

### B.4. Proof of Theorem 4.2

*Proof.* Let's introduce the dense reward reparameterzation theory in (Rafailov et al., 2024a):

**Lemma B.2.** *(Reparameterzed dense reward) (Rafailov et al., 2024a) Given a reference policy $\pi_{\text{ref}}$ and a parameter $\beta > 0$ all reward classes consistent with the Plackett-Luce (and Bradley-Terry) models, the step-wise reward $r(\boldsymbol{s}_t, \boldsymbol{a}_t)$ can be represented with the a re-parameterization*

$$
r(\boldsymbol{s}_t, \boldsymbol{a}_t) = \beta \log \pi(\boldsymbol{a}_t|\boldsymbol{s}_t) - \beta \log \pi_{\text{ref}}(\boldsymbol{a}_t|\boldsymbol{s}_t)
\tag{27}
$$

*within the token MDP where $V^*(\boldsymbol{s}_t) = 0$ for all terminal state.*

Here we combine Lemma.4.1 and Lemma.B.2 to certify our theorem. Specifically, $V^{\pi_\theta}(\boldsymbol{s}_T) = V^*(\boldsymbol{s}_T) = 0$ for all terminal state given any sequence, since $\theta$ is optimized from DPO-like algorithms. By Lemma.B.2, we have the dense reward decomposition

$$
r(\boldsymbol{s}_t^{(w)}, \hat{\boldsymbol{a}}_t) = \beta \log \pi_\theta(\hat{\boldsymbol{a}}_t|\boldsymbol{s}_t^{(w)}) - \beta \log \pi_{\text{ref}}(\hat{\boldsymbol{a}}_t|\boldsymbol{s}_t^{(w)}) = \beta \log \frac{\pi_\theta(\hat{\boldsymbol{a}}_t|\boldsymbol{s}_t^{(w)})}{\pi_{\text{ref}}(\hat{\boldsymbol{a}}_t|\boldsymbol{s}_t^{(w)})}, \forall t \in [T_w].
\tag{28}
$$

The decomposition holds in the dispreferred states such that

$$
r(\boldsymbol{s}_t^{(l)}, \check{\boldsymbol{a}}_t) = \beta \log \pi_\theta(\check{\boldsymbol{a}}_t|\boldsymbol{s}_t^{(l)}) - \beta \log \pi_{\text{ref}}(\check{\boldsymbol{a}}_t|\boldsymbol{s}_t^{(l)}) = \beta \log \frac{\pi_\theta(\check{\boldsymbol{a}}_t|\boldsymbol{s}_t^{(l)})}{\pi_{\text{ref}}(\check{\boldsymbol{a}}_t|\boldsymbol{s}_t^{(l)})}, \forall t \in [T_l].
\tag{29}
$$

where the dispreferred state-action trajectory along with the token-level MDP, *i.e.*, $\{(\boldsymbol{s}_t^{(l)}, \boldsymbol{a}_t^{(l)})\}_{t=1}^{T_l}$ is constructed by

$s_t^{(l)} = (x, y_{<t}^{(l)})$ and $a_t^{(l)} = y_t^{(l)}, \forall t \in [T_l]$. Given this, we have

$$U_{\text{ss}}(x, y^{(w)}, y^{(l)}) = R(x, y^{(w)}) - R(x, y^{(l)})$$

$$= \left( R(x, y^{(w)}) - \sum_{t=1}^{T_w} r(s_t^{(w)}, \hat{a}_t) \right) + \left( \sum_{t=1}^{T_w} r(s_t^{(w)}, \hat{a}_t) - \sum_{t=1}^{T_l} r(s_t^{(l)}, \check{a}_t) \right) + \left( \sum_{t=1}^{T_l} r(s_t^{(l)}, \check{a}_t) - R(x, y^{(l)}) \right)$$

$$= \underbrace{\sum_{t=1}^{T_w} -\lambda_N(s_t^{(w)}) \left( \frac{\pi_\theta(a_t^{(w)}|s_t^{(w)})}{\overline{\pi}_\varphi(a_t^{(w)}|s_t^{(w)})} - \frac{\pi_\theta(\hat{a}_t|s_t^{(w)})}{\overline{\pi}_\varphi(\hat{a}_t|s_t^{(w)})} \right)}_{\mu_w(\varphi;\theta)} + \underbrace{\sum_{t=1}^{T_l} \lambda_N(s_t^{(l)}) \left( \frac{\pi_\theta(a_t^{(l)}|s_t^{(l)})}{\overline{\pi}_\varphi(a_t^{(l)}|s_t^{(l)})} - \frac{\pi_\theta(\check{a}_t|s_t^{(l)})}{\overline{\pi}_\varphi(\check{a}_t|s_t^{(l)})} \right)}_{\mu_l(\varphi;\theta)}$$

$$+ \underbrace{\sum_{t=1}^{T_w} \beta \log \frac{\pi_\theta(\hat{a}_t|s_t^{(w)})}{\pi_{\text{ref}}(\hat{a}_t|s_t^{(w)})} - \sum_{t=1}^{T_l} \beta \log \frac{\pi_\theta(\check{a}_t|s_t^{(l)})}{\pi_{\text{ref}}(\check{a}_t|s_t^{(l)})}}_{\delta_\theta}$$

(30)

The theorem has been proved. □

## B.5. Proof of Theorem.4.3

Here we provide the formal proofs to the sentence-level version of Lemma.4.1 and Theorem.3.3. They are based on the definition of Sentence-level MDP:

**Definition B.3. (Sentence-level MDP)** Suppose that $\mathcal{M} = (\mathcal{S}, \mathcal{A}, \mathcal{P}, \mathcal{R}, \mathcal{T})$ denotes the token-level MDP. It identifies the sentence-level MDP $\mathcal{M}_S = (\mathcal{S}_S, \mathcal{A}_S, \mathcal{P}_S, \mathcal{R}_S, \mathcal{T}_S)$ as

$$\mathcal{S}_S = \bigcup_{n=1}^{+\infty} \mathcal{S}^n; \mathcal{A}_S = \mathcal{A}_{\text{MCTS}} = \bigcup_{n=1}^{N} \mathcal{A}^n; \mathcal{P}_S : \mathcal{S}_S \times \mathcal{A}_S \to \mathcal{S}_S; \mathcal{R}_S : \mathcal{S}_S \times \mathcal{A}_S \to \mathbb{R}; \mathcal{T}_S = \mathcal{T}. \quad (31)$$

where $\mathcal{S}^n$ indicates the sequence composed of $n$ sentences ended by "\n" in order and obviously, $\mathcal{S}^n \subset \mathcal{S}$. $\mathcal{A}^n$ is the sentence-level action space with $n$-length tokens and $N$ indicates the maximum length of sentences. The transition kernel $\mathcal{P}_S$ and reward space $\mathcal{R}_S$ holds $\mathcal{P}_S \subset \mathcal{S} \times \mathcal{A}_S \to \mathcal{S}$ and $\mathcal{R}_S \subset \mathcal{S} \times \mathcal{A}_S \to \mathbb{R}$. To this, if we consider the policies hold the decomposition $\pi^S(A_t|s_t) = \prod_{i=1}^{N_t} \pi(a_t^{(i)}|s_t^{(i)})$ ($A_t = (a_t^{(i)})_{i=1}^{N_t} \in \mathcal{A}_S$ denotes the sentence-level action, $N_t$ indicates the token number in the selected sentence action, $a_t^{(i)}$ indicates the $i$-th token in the $t$-th sentence, and $s^{(i)} = (s, a_t^{(1)}, \cdots, a_t^{(i-1)})$.) $\mathcal{R}_S \subset \mathcal{S} \times \mathcal{A}_S \to \mathbb{R}$, to this, we define $r^S(A_i, s) \in \mathcal{R}_S$ such that $r^S(s_t, A_i) = \sum_i^{N_t} r(s_t^{(i)}, a_t^{(i)})$.

**Lemma B.4.** Given the prompt $x$ with its preference response pair $y^{(w)}, y^{(l)}$ composed of $T_w^S, T_l^S$ sentences, $A_t^{(w)}, A_t^{(l)}$ denotes the $t$-th sentence in $y^{(w)}, y^{(l)}$, respectively. Suppose the state $s_t^{(w)}, s_t^{(l)}$ transmits along sentence-level MDP, $a_t^{(w,i)}/a_t^{(l,i)}$ indicates the $i$-th token in the $A_t^{(w)}/A_t^{(l)}, s_t^{(w,i)}/s_t^{(l,i)}$ denotes the sequential context ahead of $a_t^{(w,i)}/a_t^{(l,i)}$ in $y^{(w)}/y^{(l)}$. Suppose that the sentence-level policies $\pi_{\text{ref}}^S, \pi_\theta^S, \overline{\pi}_\varphi^S$ are identified by LM-based token-selection policies $\pi_{\text{ref}}, \pi_\theta, \overline{\pi}_\varphi$, respectively; $\hat{A}_t = (\hat{a}_t^{(1)}, \cdots, \hat{a}_t^{(\hat{N}_t)}) \sim \overline{\pi}_\varphi^S(\cdot|s_t^{(w)}), \hat{s}_t^{(0)} = s_t^{(w)}, \hat{s}_t^{(i+1)} = \mathcal{P}(\hat{s}_t^{(i)}, \hat{a}_t^{(i)}); \check{A}_t = (\check{a}_t^{(1)}, \cdots, \check{a}_t^{(\check{N}_t)}) \sim \overline{\pi}_\varphi^S(\cdot|s_t^{(l)}), \check{s}_t^{(0)} = s_t^{(l)}, \check{s}_t^{(i+1)} = \mathcal{P}(\check{s}_t^{(i)}, \check{a}_t^{(i)})$. It holds

$$R(x, y^{(w)}) = \sum_{t=1}^{T_w^S} \left( \sum_{i=1}^{\hat{N}_t} r(\hat{s}_t^{(i)}, \hat{a}_t^{(i)}) - \lambda_N(s_t^{(w)}) \left( \prod_{i=1}^{|A_t^{(w)}|} \frac{\pi_\theta(a_t^{(w,i)}|s_t^{(w,i)})}{\overline{\pi}_\varphi(a_t^{(w,i)}|s_t^{(w,i)})} - \prod_{i=1}^{\hat{N}_t} \frac{\pi_\theta(\hat{a}_t^{(i)}|\hat{s}_t^{(i)})}{\overline{\pi}_\varphi(\hat{a}_t^{(i)}|\hat{s}_t^{(i)})} \right) \right)$$

$$R(x, y^{(l)}) = \sum_{t=1}^{T_l^S} \left( \sum_{i=1}^{\check{N}_t} r(\check{s}_t^{(i)}, \check{a}_t^{(i)}) - \lambda_N(s_t^{(l)}) \left( \prod_{i=1}^{|A_t^{(l)}|} \frac{\pi_\theta(a_t^{(l,i)}|s_t^{(l,i)})}{\overline{\pi}_\varphi(a_t^{(l,i)}|s_t^{(l,i)})} - \prod_{i=1}^{\check{N}_t} \frac{\pi_\theta(\check{a}_t^{(i)}|\check{s}_t^{(i)})}{\overline{\pi}_\varphi(\check{a}_t^{(i)}|\check{s}_t^{(i)})} \right) \right)$$

(32)

*Proof.* Suppose that $Q^{\pi_\theta^S}(s_t, \cdot)$ denotes the state-action value function on the sentence-level policy $\pi_\theta^S$ with respect to LM $\theta$. Due to $\pi_\theta^S$ defined with the action space $\mathcal{A}_S = \mathcal{A}_{\text{MCTS}}$, it results in

$$\overline{\pi}_\varphi^S(A_t|s_t) = \lambda_N(s_t) \frac{\pi_\theta^S(A_t|s_t)}{\alpha(s_t) - Q^{\pi_\theta^S}(s_t, A_t)} \leftrightarrows r^S(s_t, A_t) = \alpha(s_t) - V^{\pi_\theta^S}(s_t) - \lambda_N(s_t) \frac{\pi_\theta^S(A_t|s_t)}{\overline{\pi}_\varphi^S(A_t|s_t)}, \quad (33)$$

where $A_t = (a_t^{(i)})_{i=1}^{N_t} \in \mathcal{A}_S$ denotes the sentence-level action, $N_t$ indicates the token number in the selected sentence action, and $a_t^{(i)}$ indicates the $i$-th token in the $t$-th sentence. It holds $Q^{\pi_\theta^S}(\boldsymbol{s}_t, A_t) = r^S(\boldsymbol{s}_t, A_t) + V^{\pi_\theta}(\boldsymbol{s}_t) = \sum_i^{N_t} r(\boldsymbol{s}_t^{(i)}, a_t^{(i)}) + V^{\pi_\theta^S}(\boldsymbol{s}_t)$.

Provided $(\boldsymbol{x}, \boldsymbol{y}^{(w)})$ denoting a pair of prompt and its preferred response, we construct the preferred state-action trajectory along with the sentence-level MDP. In particular, suppose $T_w^S$ denotes the number of sentences included in the preferred response $\boldsymbol{y}^{(w)}$, which can be uniquely identified through the location of "$\backslash n$" tokens. We denote them as a sequence of sentence-level actions $\{A_t^{(w)}\}_{t=1}^{T_w^S}$. $\forall t \in T_w^S$, and make the decomposition such that $\{(\boldsymbol{s}_t^{(w)}, A_t^{(w)})\}_{t=1}^{T_w^S}$ with respect to

$$
\begin{aligned}
A_t^{(w)} &= (y^{(w)}_{\sum_{j=1}^{t-1} N_j^{(w)}+1}, \cdots, y^{(w)}_{\sum_{j=1}^{t-1} N_j^{(w)}+N_t^{(w)}}) \\
&= (\boldsymbol{a}_t^{(w,1)}, \cdots, \boldsymbol{a}_t^{(w,N_t^{(w)})}) \\
\boldsymbol{s}_t^{(w)} &= (\boldsymbol{x}, A_1^{(w)}, \cdots, A_{t-1}^{(w)}),
\end{aligned}
\tag{34}
$$

Then for each $(\boldsymbol{s}_t^{(w)}, A_t^{(w)})$

$$
\boldsymbol{s}_t^{(w,i)} = (\boldsymbol{s}_t^{(w)}, \boldsymbol{a}_t^{(w,1)}, \cdots, \boldsymbol{a}_t^{(w,i-1)}), \forall i \in [N_t^{(w)}].
\tag{35}
$$

where $\boldsymbol{a}_t^{(w,i)}$ denotes the response token $y^{(w)}_{\sum_{j=1}^{t-1} N_j^{(w)}+i}$ in $\boldsymbol{y}^{(w)}$.

Then we provide the derivation

$$
\begin{aligned}
R(\boldsymbol{x}, \boldsymbol{y}^{(w)}) &= \sum_{t=1}^{T_w^S} \sum_i^{N_t^{(w)}} r(\boldsymbol{s}_t^{(i)}, a_t^{(i)}) = \sum_{t=1}^{T_w^S} r^S(\boldsymbol{s}_t^{(w)}, A_t) \\
&= \sum_{t=1}^{T_w^S} \left( \alpha(\boldsymbol{s}_t^{(w)}) - V^{\pi_\theta^S}(\boldsymbol{s}_t^{(w)}) - \lambda_N(\boldsymbol{s}_t^{(w)}) \frac{\pi_\theta^S(A_t^{(w)}|\boldsymbol{s}_t^{(w)})}{\overline{\pi}_\varphi^S(A_t^{(w)}|\boldsymbol{s}_t^{(w)})} \right) \\
&= \sum_{t=1}^{T_w^S} \left( \alpha(\boldsymbol{s}_t^{(w)}) - V^{\pi_\theta}(\boldsymbol{s}_t^{(w)}) - \lambda_N(\boldsymbol{s}_t^{(w)}) \frac{\prod_{i=1}^{N_t^{(w)}} \pi_\theta(\boldsymbol{a}_t^{(w,i)}|\boldsymbol{s}_t^{(w,i)})}{\prod_{i=1}^{N_t^{(w)}} \overline{\pi}_\varphi(\boldsymbol{a}_t^{(w,i)}|\boldsymbol{s}_t^{(w,i)})} \right) \\
&= \sum_{t=1}^{T_w^S} \left( \alpha(\boldsymbol{s}_t^{(w)}) - V^{\pi_\theta}(\boldsymbol{s}_t^{(w)}) - \lambda_N(\boldsymbol{s}_t^{(w)}) \prod_{i=1}^{N_t^{(w)}} \frac{\pi_\theta(\boldsymbol{a}_t^{(w,i)}|\boldsymbol{s}_t^{(w,i)})}{\overline{\pi}_\varphi(\boldsymbol{a}_t^{(w,i)}|\boldsymbol{s}_t^{(w,i)})} \right)
\end{aligned}
\tag{36}
$$

Beyond this, for the $t$-th state in the preferred state-action trajectory, we sample another action $\hat{A}_t$ from the parameterized ITS exploration policy $\overline{\pi}_\varphi^S$ i.e., $\hat{A}_t = (\hat{\boldsymbol{a}}_t^1, \cdots, \hat{\boldsymbol{a}}_t^{(\hat{N}_t)}) \sim \overline{\pi}_\varphi^S(\cdot|\boldsymbol{s}_t^{(w)})$. It can be telescoped into $\{\hat{\boldsymbol{a}}_t^{(i)} \sim \overline{\pi}_\varphi(\cdot|\hat{\boldsymbol{s}}_t^{(i)})\}_{i=1}^{\hat{N}_t}$ with respect to $\hat{\boldsymbol{s}}_t^{(i)} = (\boldsymbol{x}, A_1^{(w)}, \cdots, A_{t-1}^{(w)}, \hat{\boldsymbol{a}}_t^{(1)}, \cdots, \hat{\boldsymbol{a}}_t^{(i-1)})$. Therefore $\forall t \in [T_w^S]$, it holds

$$
\begin{aligned}
r^S(\boldsymbol{s}_t^{(w)}, \hat{A}_t) = \sum_{i=1}^{N_t} r(\hat{\boldsymbol{a}}_t^{(i)}, \hat{\boldsymbol{s}}_t^{(i)}) &= \alpha(\boldsymbol{s}_t^{(w)}) - V^{\pi_\theta^S}(\boldsymbol{s}_t^{(w)}) - \lambda_N(\boldsymbol{s}_t^{(w)}) \frac{\pi_\theta^S(\hat{A}_t|\boldsymbol{s}_t^{(w)})}{\overline{\pi}_\varphi^S(\hat{A}_t|\boldsymbol{s}_t^{(w)})} \\
&= \alpha(\boldsymbol{s}_t^{(w)}) - V^{\pi_\theta^S}(\boldsymbol{s}_t^{(w)}) - \lambda_N(\boldsymbol{s}_t^{(w)}) \frac{\prod_{i=1}^{\hat{N}_t} \pi_\theta(\hat{\boldsymbol{a}}_t^{(i)}|\hat{\boldsymbol{s}}_t^{(i)})}{\prod_{i=1}^{\hat{N}_t} \overline{\pi}_\varphi(\hat{\boldsymbol{a}}_t^{(i)}|\hat{\boldsymbol{s}}_t^{(i)})} \\
&= \alpha(\boldsymbol{s}_t^{(w)}) - V^{\pi_\theta^S}(\boldsymbol{s}_t^{(w)}) - \lambda_N(\boldsymbol{s}_t^{(w)}) \prod_{i=1}^{\hat{N}_t} \frac{\pi_\theta(\hat{\boldsymbol{a}}_t^{(i)}|\hat{\boldsymbol{s}}_t^{(i)})}{\overline{\pi}_\varphi(\hat{\boldsymbol{a}}_t^{(i)}|\hat{\boldsymbol{s}}_t^{(i)})}
\end{aligned}
\tag{37}
$$

Derived from the similar routine in Eq.26, we combine Eq.36, 37 to obtain

$$
R(\boldsymbol{x}, \boldsymbol{y}^{(w)}) = \sum_{t=1}^{T_w^S} \left( \sum_{i=1}^{\hat{N}_t} r(\hat{\boldsymbol{s}}_t^{(i)}, \hat{\boldsymbol{a}}_t^{(i)}) - \lambda_N(\boldsymbol{s}_t^{(w)}) \left( \prod_{i=1}^{N_t^{(w)}} \frac{\pi_\theta(\boldsymbol{a}_t^{(w,i)}|\boldsymbol{s}_t^{(w,i)})}{\overline{\pi}_\varphi(\boldsymbol{a}_t^{(w,i)}|\boldsymbol{s}_t^{(w,i)})} - \prod_{i=1}^{\hat{N}_t} \frac{\pi_\theta(\hat{\boldsymbol{a}}_t^{(i)}|\hat{\boldsymbol{s}}_t^{(i)})}{\overline{\pi}_\varphi(\hat{\boldsymbol{a}}_t^{(i)}|\hat{\boldsymbol{s}}_t^{(i)})} \right) \right)
\tag{38}
$$

For the dispreferred response $\boldsymbol{y}^{(l)}$, it refers to the mirror notation of $(w) \rightarrow (l)$ to construct the dispreferred state-action trajectory $\{(\boldsymbol{s}_t^{(l)}, A_t^{(l)})\}_{t=1}^{T_l^S}$ and for each $(\boldsymbol{s}_t^{(l)}, A_t^{(l)})$, $\boldsymbol{a}_t^{(l,i)} = y_{\sum_{j=1}^{t-1} N_j^{(l)}+i}^{(w)}$ and $\boldsymbol{s}_t^{(l,i)} = (\boldsymbol{s}_t^{(l)}, \boldsymbol{a}_t^{(l,1)}, \cdots, \boldsymbol{a}_t^{(l,i-1)}), \forall i \in$ $[N_t^{(l)}]$ in $\boldsymbol{y}^{(l)}$. Given this, for the $t$-th state in the dispreferred state-action trajectory, instead of $A_t^{(l)}$, we sample another sentence action $\check{A}_t$ from the parameterized ITS exploration policy $\overline{\pi}_\varphi^S$ i.e., $\check{A}_t = (\hat{\boldsymbol{a}}_t^1, \cdots, \check{\boldsymbol{a}}_t^{(\check{N}_t)}) \sim \overline{\pi}_\varphi^S(\cdot|\boldsymbol{s}_t^{(l)})$. It can be telescoped into $\{\check{\boldsymbol{a}}_t^{(i)} \sim \overline{\pi}_\varphi(\cdot|\check{\boldsymbol{s}}_t^{(i)})\}_{i=1}^{\check{N}_t}$ with respect to $\check{\boldsymbol{s}}_t^{(i)} = (\boldsymbol{x}, A_1^{(l)}, \cdots, A_{t-1}^{(l)}, \check{\boldsymbol{a}}_t^{(1)}, \cdots, \check{\boldsymbol{a}}_t^{(i-1)})$. From the same deduction routine, it holds

$$r^S(\boldsymbol{s}_t^{(l)}, \check{A}_t) = \alpha(\boldsymbol{s}_t^{(l)}) - V^{\pi_\theta^S}(\boldsymbol{s}_t^{(l)}) - \lambda_N(\boldsymbol{s}_t^{(l)}) \prod_{i=1}^{\check{N}_t} \frac{\pi_\theta(\check{\boldsymbol{a}}_t^{(i)}|\check{\boldsymbol{s}}_t^{(i)})}{\overline{\pi}_\varphi(\check{\boldsymbol{a}}_t^{(i)}|\check{\boldsymbol{s}}_t^{(i)})}, \forall t \in [T_l^S] \tag{39}$$

and

$$R(\boldsymbol{x}, \boldsymbol{y}^{(l)}) = \sum_{t=1}^{T_l^S} \left( \sum_{i=1}^{\check{N}_t} r(\check{\boldsymbol{s}}_t^{(i)}, \check{\boldsymbol{a}}_t^{(i)}) - \lambda_N(\boldsymbol{s}_t^{(l)}) \left( \prod_{i=1}^{N_t^{(l)}} \frac{\pi_\theta(\boldsymbol{a}_t^{(l,i)}|\boldsymbol{s}_t^{(l,i)})}{\overline{\pi}_\varphi(\boldsymbol{a}_t^{(l,i)}|\boldsymbol{s}_t^{(l,i)})} - \prod_{i=1}^{\check{N}_t} \frac{\pi_\theta(\check{\boldsymbol{a}}_t^{(i)}|\check{\boldsymbol{s}}_t^{(i)})}{\overline{\pi}_\varphi(\check{\boldsymbol{a}}_t^{(i)}|\check{\boldsymbol{s}}_t^{(i)})} \right) \right) \tag{40}$$

$\square$

On top of Lemma.B.4, we prove Theorem 4.3 as follows

*Proof.* According to Lemm.B.2, we have

$$r(\hat{\boldsymbol{s}}_t^{(i)}, \hat{\boldsymbol{a}}_t^{(i)}) = \beta \log \frac{\pi_\theta(\hat{\boldsymbol{a}}_t^{(i)}|\hat{\boldsymbol{s}}_t^{(i)})}{\pi_{\text{ref}}(\hat{\boldsymbol{a}}_t^{(i)}|\hat{\boldsymbol{s}}_t^{(i)})}, \text{s.t.} \forall t \in T_w^S, \forall i \in \hat{N}_t;$$

$$r(\check{\boldsymbol{s}}_t^{(i)}, \check{\boldsymbol{a}}_t^{(i)}) = \beta \log \frac{\pi_\theta(\check{\boldsymbol{a}}_t^{(i)}|\check{\boldsymbol{s}}_t^{(i)})}{\pi_{\text{ref}}(\check{\boldsymbol{a}}_t^{(i)}|\check{\boldsymbol{s}}_t^{(i)})}, \text{s.t.} \forall t \in T_l^S, \forall i \in \check{N}_t. \tag{41}$$

Therefore

$$U_{\text{sa}}(\boldsymbol{x}, \boldsymbol{y}^{(w)}, \boldsymbol{y}^{(l)}) = R(\boldsymbol{x}, \boldsymbol{y}^{(w)}) - R(\boldsymbol{x}, \boldsymbol{y}^{(l)})$$

$$= \left( R(\boldsymbol{x}, \boldsymbol{y}^{(w)}) - \sum_{t=1}^{T_w^S}\sum_{i=1}^{\hat{N}_t} r(\hat{\boldsymbol{s}}_t^{(i)}, \hat{\boldsymbol{a}}_t^{(i)}) \right) + \left( \sum_{t=1}^{T_w^S}\sum_{i=1}^{\hat{N}_t} r(\hat{\boldsymbol{s}}_t^{(i)}, \hat{\boldsymbol{a}}_t^{(i)}) - \sum_{t=1}^{T_l^S}\sum_{i=1}^{\check{N}_t} r(\check{\boldsymbol{s}}_t^{(i)}, \check{\boldsymbol{a}}_t^{(i)}) \right) + \left( \sum_{t=1}^{T_l^S}\sum_{i=1}^{\check{N}_t} r(\check{\boldsymbol{s}}_t^{(i)}, \check{\boldsymbol{a}}_t^{(i)}) - R(\boldsymbol{x}, \boldsymbol{y}^{(l)}) \right)$$

$$= \left( R(\boldsymbol{x}, \boldsymbol{y}^{(w)}) - \sum_{t=1}^{T_w^S}\sum_{i=1}^{\hat{N}_t} r(\hat{\boldsymbol{s}}_t^{(i)}, \hat{\boldsymbol{a}}_t^{(i)}) \right) + \left( R(\boldsymbol{x}, \boldsymbol{y}^{(l)}) - \sum_{t=1}^{T_l^S}\sum_{i=1}^{\check{N}_t} r(\check{\boldsymbol{s}}_t^{(i)}, \check{\boldsymbol{a}}_t^{(i)}) \right)$$

$$+ \beta \left( \sum_{t=1}^{T_w^S}\sum_{i=1}^{\hat{N}_t} \log \frac{\pi_\theta(\hat{\boldsymbol{a}}_t^{(i)}|\hat{\boldsymbol{s}}_t^{(i)})}{\pi_{\text{ref}}(\hat{\boldsymbol{a}}_t^{(i)}|\hat{\boldsymbol{s}}_t^{(i)})} - \sum_{t=1}^{T_l^S}\sum_{i=1}^{\check{N}_t} \log \frac{\pi_\theta(\check{\boldsymbol{a}}_t^{(i)}|\check{\boldsymbol{s}}_t^{(i)})}{\pi_{\text{ref}}(\check{\boldsymbol{a}}_t^{(i)}|\check{\boldsymbol{s}}_t^{(i)})} \right)$$

$$= \underbrace{\sum_{t=1}^{T_w^S} -\lambda_N(\boldsymbol{s}_t^{(w)}) \left( \prod_{i=1}^{N_t^{(w)}} \frac{\pi_\theta(\boldsymbol{a}_t^{(w,i)}|\boldsymbol{s}_t^{(w,i)})}{\overline{\pi}_\varphi(\boldsymbol{a}_t^{(w,i)}|\boldsymbol{s}_t^{(w,i)})} - \prod_{i=1}^{\hat{N}_t} \frac{\pi_\theta(\hat{\boldsymbol{a}}_t^{(i)}|\hat{\boldsymbol{s}}_t^{(i)})}{\overline{\pi}_\varphi(\hat{\boldsymbol{a}}_t^{(i)}|\hat{\boldsymbol{s}}_t^{(i)})} \right)}_{\mu_w^S(\varphi, \theta)} + \underbrace{\sum_{t=1}^{T_l^S} \lambda_N(\boldsymbol{s}_t^{(l)}) \left( \prod_{i=1}^{N_t^{(l)}} \frac{\pi_\theta(\boldsymbol{a}_t^{(l,i)}|\boldsymbol{s}_t^{(l,i)})}{\overline{\pi}_\varphi(\boldsymbol{a}_t^{(l,i)}|\boldsymbol{s}_t^{(l,i)})} - \prod_{i=1}^{\check{N}_t} \frac{\pi_\theta(\check{\boldsymbol{a}}_t^{(i)}|\check{\boldsymbol{s}}_t^{(i)})}{\overline{\pi}_\varphi(\check{\boldsymbol{a}}_t^{(i)}|\check{\boldsymbol{s}}_t^{(i)})} \right)}_{-\mu_l^S(\varphi, \theta)}$$

$$+ \underbrace{\beta \left( \sum_{t=1}^{T_w^S}\sum_{i=1}^{\hat{N}_t} \log \frac{\pi_\theta(\hat{\boldsymbol{a}}_t^{(i)}|\hat{\boldsymbol{s}}_t^{(i)})}{\pi_{\text{ref}}(\hat{\boldsymbol{a}}_t^{(i)}|\hat{\boldsymbol{s}}_t^{(i)})} - \sum_{t=1}^{T_l^S}\sum_{i=1}^{\check{N}_t} \log \frac{\pi_\theta(\check{\boldsymbol{a}}_t^{(i)}|\check{\boldsymbol{s}}_t^{(i)})}{\pi_{\text{ref}}(\check{\boldsymbol{a}}_t^{(i)}|\check{\boldsymbol{s}}_t^{(i)})} \right)}_{\delta^S(\theta)}$$

$$\tag{42}$$

Set up $N_i^{(w)} = |A_t^{(w)}|$ and $N_i^{(l)} = |A_t^{(l)}|$, then the theorem has been proved. $\square$

---

**Algorithm 1** The algorithm pipeline of IT-PO

---

**Input:** preference dataset $\mathcal{D}$; pre-trained LLMs $\theta, \varphi$.

**Hyper-parameters:** $\epsilon$; $K$; the batch size $M$ to construct the batch of cDPO; the number of alternative training $n_{\text{alter}}$.

**Output:** LLMs $\theta^*, \varphi^*$.

Initialize LLMs $\theta, \varphi$ by SF with the prompt-response pairs drawn from $\mathcal{D}$, $N_{\text{alter}} = 0$;

Minimize $\mathcal{L}_{\text{cDPO}}(\theta)$ to post-train LLM $\theta$ with the prompt and its pairwise responses drawn from the dataset $\mathcal{D}$;

**repeat**

  **Implicit-Tree Preference Optimization:**

  **repeat**

    Construct the training batch by $M$ triplets drawn from $\mathcal{D}$;

    For each triplet $(\boldsymbol{x}, \boldsymbol{y}^{(w)}, \boldsymbol{y}^{(l)})$, construct $\{(\boldsymbol{a}_t^{(w)}, \boldsymbol{s}_t^{(w)})\}_{t=1}^{T_w}, \{(\boldsymbol{a}_t^{(l)}, \boldsymbol{s}_t^{(l)})\}_{t=1}^{T_l}$ based on the rule of token-level MDP (*w.r.t.* $\mathcal{L}_{\text{ss-IT-PO}}(\varphi; \theta)$ for LM alignment) or sentence-level MDP (*w.r.t.* $\mathcal{L}_{\text{as-IT-PO}}(\varphi; \theta)$ for LM Reasoning);

    Set $K_w = \frac{K}{T_w}, \forall t \in T_w$ (**parallel**), draw $K_w$ tokens from $\overline{\pi}_\varphi(\cdot|\boldsymbol{s}_t^{(w)})$ to construct the samples of token-level action random variable $\hat{\boldsymbol{a}}_t$, or draw $K_w$ sentences from $\overline{\pi}_\varphi^S(\cdot|\boldsymbol{s}_t^{(w)})$ to construct the samples of the sentence-level action random variable $\hat{A}_t$;

    Set $K_l = \frac{K}{T_l}, \forall t \in T_l$ (**parallel**), draw $K_l$ tokens from $\overline{\pi}_\varphi(\cdot|\boldsymbol{s}_t^{(l)})$ to construct the samples of token-level action random variable $\check{\boldsymbol{a}}_t$, or draw $K_l$ sentences from $\overline{\pi}_\varphi^S(\cdot|\boldsymbol{s}_t^{(l)})$ to construct the samples of the sentence-level action random variable $\check{A}_t$;

    Fix $\theta$, minimize $\mathcal{L}_{\text{ss-IT-PO}}(\varphi; \theta)$ to update $\varphi$ for LM alignment or minimize $\mathcal{L}_{\text{as-IT-PO}}(\varphi; \theta)$ to update $\varphi$ for LM reasoning;

  **until** The number of training epoches reach the same of $\mathcal{L}_{\text{cDPO}}(\theta)$.

  **Preference Policy Augmentation:**

  For each prompt $\boldsymbol{x}$ in $\mathcal{D}$, generate $K$ responses $\{\boldsymbol{y}_k\}_{i=1}^K$ using $\overline{\pi}_\varphi$;

  For each prompt $\boldsymbol{x}$ in $\mathcal{D}$ and $\forall i \neq j \in [K]$, calculate $U_{\text{ss}}(\boldsymbol{x}, \boldsymbol{y}_i, \boldsymbol{y}_j)$ or $U_{\text{as}}(\boldsymbol{x}, \boldsymbol{y}_i, \boldsymbol{y}_j)$;

  Build the token-level policy augmentation pool $\mathcal{D}_{\text{ss}}^+ = \{(\boldsymbol{x}, \boldsymbol{y}_i, \boldsymbol{y}_j) | \forall i \neq j \in [K], U_{\text{ss}}(\boldsymbol{x}, \boldsymbol{y}_i, \boldsymbol{y}_j) > 0, (\boldsymbol{x}, \boldsymbol{y}_i, \boldsymbol{y}_j) \in \text{top}_K(\{U_{\text{ss}}(\boldsymbol{x}, \boldsymbol{y}_i, \boldsymbol{y}_j)\}_{i \neq j \in [K]})$, or build the sentence-level policy augmentation pool $\mathcal{D}_{\text{as}}^+ = \{(\boldsymbol{x}, \boldsymbol{y}_i, \boldsymbol{y}_j) | \forall i \neq j \in [K], U_{\text{as}}(\boldsymbol{x}, \boldsymbol{y}_i, \boldsymbol{y}_j) > 0, (\boldsymbol{x}, \boldsymbol{y}_i, \boldsymbol{y}_j) \in \text{top}_K(\{U_{\text{as}}(\boldsymbol{x}, \boldsymbol{y}_i, \boldsymbol{y}_j)\}_{i \neq j \in [K]})$

  Minimize $\mathcal{L}_{\text{cDPO}}(\theta)$ to post-train LLM $\theta$ with the prompt and its pairwise responses drawn from $\mathcal{D} \cup \mathcal{D}_{\text{ss}}^+$ or $\mathcal{D} \cup \mathcal{D}_{\text{as}}^+$;

  $N_{\text{alter}} = N_{\text{alter}} + 1$;

**until** $N_{\text{alter}} = n_{\text{alter}}$

$\theta^* \leftarrow \theta, \varphi^* \leftarrow \varphi$.

---

## C. Implementation

The algorithm pipeline of IT-PO are presented in Algorithm.1. The algorithm implementation almost holds the consistency with its theoretical foundation except for two details:(1) the specification of $\lambda_N(\boldsymbol{s}_t)$; (2) the $\varphi$'s gradient implementation in $\mathcal{L}_{\text{ss-IT-PO}}(\varphi; \theta)$ and $\mathcal{L}_{\text{as-IT-PO}}(\varphi; \theta)$.

Suppose that $\lambda_N(\boldsymbol{s}_t^{(w)}), \lambda_N(\boldsymbol{s}_t^{(l)})$ denote the number of reasoning paths generated from the state $\boldsymbol{s}_t$, which consists of a prompt $\boldsymbol{x}$ and its responses $\boldsymbol{y}^{(w)}, \boldsymbol{y}^{(l)}$ with their contents in the previous $t-1$ steps. For each preference pair, we set $\forall t \in \{1, \ldots, T_w\}, \lambda_N(\boldsymbol{s}_t^{(w)}) = K_w$ and $\forall t \in \{1, \ldots, T_l\}, \lambda_N(\boldsymbol{s}_t^{(l)}) = K_l$, where $K_w, K_l$ denote how many search start from the $t$-th leaf nodes $\boldsymbol{s}_t^{(w)}, \boldsymbol{s}_t^{(l)}$, respectively. Notice that $T_w, T_l$ change with respect to the preference pair. So IT-PO adaptively configure $K_w = \frac{K}{T_w}, K_l = \frac{K}{T_l}$ to balance the optimization with different response lengths in $\boldsymbol{y}^{(w)}, \boldsymbol{y}^{(l)}$ for each pair. We set $K = 8$, inspired from the number of sampled responses for each prompt in many RLHF implementations.

In $\varphi$'s gradient analysis in $\mathcal{L}_{\text{ss-IT-PO}}(\varphi; \theta)$, the gradient of $\varphi$ consists of terms in the form $\frac{\pi_\theta(\boldsymbol{a}_t|\boldsymbol{s}_t)}{\overline{\pi}_\varphi(\boldsymbol{a}_t|\boldsymbol{s}_t)} \nabla \log[\overline{\pi}(\boldsymbol{a}_t|\boldsymbol{s}_t)]$. Due to $\frac{\pi_\theta(\boldsymbol{a}_t|\boldsymbol{s}_t)}{\overline{\pi}_\varphi(\boldsymbol{a}_t|\boldsymbol{s}_t)} \in (0, +\infty)$, updating the models with $\frac{\pi_\theta(\boldsymbol{a}_t|\boldsymbol{s}_t)}{\overline{\pi}_\varphi(\boldsymbol{a}_t|\boldsymbol{s}_t)} \nabla \log[\overline{\pi}(\boldsymbol{a}_t|\boldsymbol{s}_t)]$ may suffer from exploding/vanishing gradients. To this, we used their logarithmic scaling $\frac{\pi_\theta(\boldsymbol{a}_t|\boldsymbol{s}_t)}{\overline{\pi}_\varphi(\boldsymbol{a}_t|\boldsymbol{s}_t)} \rightarrow \exp(\log(\pi_\theta(\boldsymbol{a}_t|\boldsymbol{s}_t)) - \log(\overline{\pi}_\varphi(\boldsymbol{a}_t|\boldsymbol{s}_t)))$ to ensure the less sensitive

*Table 5.* Task setups. The node, tree max width, and tree max depth are search space parameters. The max tree-width and tree-depth follow the empirical experience in (Feng et al., 2023).

| Task | Task Category | Train/test size | Node | Tree Max width | Tree Max depth |
|------|---------------|-----------------|------|----------------|----------------|
| **GSM8k** | Mathematical Reasoning | 7.5k / 1.3k | Sentence | 6 | 8 |
| **Game24** | Mathematical Planning | 1.0k / 0.3k | Sentence | 20 | 4 |
| **MATH** | Mathematical Reasoning | - / - | Sentence | 6 | 8 |

*Table 6.* Performance Comparison on ProofWriter and Chess Endgame Tasks

| Setting | Baselines | ProofWriter (Acc %) | Chess Endgame (Win rate %) |
|---------|-----------|---------------------|----------------------------|
| Path@1 | CoT-greedy | 37.72 | 58.14 |
| | BFS-V | 48.94 | 67.75 |
| | MCTS-$\alpha$ | 66.71 | 96.90 |
| | MCTS-rollout | 69.23 | 98.76 |
| | ITS-$\alpha$ (ours) | 71.77 | 99.21 |
| | ITS-rollout (ours) | 75.31 | 99.83 |
| Equal-Token | CoT-SC-MAJ | 36.50 | 9.84 |
| | CoT-SC-MAJ | 36.58 | 73.80 |
| | BFS-V-ORM | 63.42 | 93.18 |
| | MCTS-ORM | 60.86 | 94.26 |
| | ITS-$\alpha$ (ours) | 74.26 | 96.48 |
| | ITS-rollout (ours) | 78.15 | 98.57 |

update ratio. In $\varphi$'s gradient analysis in $\mathcal{L}_{\text{as-IT-PO}}(\varphi; \theta)$, it can derive the similar gradient as

$$
\nabla_\varphi \mu_w^S(\varphi, \theta) = \sum_{t=1}^{T_w^S} \lambda_N(\boldsymbol{s}_t^{(w)}) \Bigg( \prod_{i=1}^{|A_t^{(w)}|} \frac{\pi_\theta(\boldsymbol{a}_t^{(w,i)}|\boldsymbol{s}_t^{(w,i)})}{\overline{\pi}_\varphi(\boldsymbol{a}_t^{(w,i)}|\boldsymbol{s}_t^{(w,i)})} \Big( \sum_{j=1}^{|A_t^{(w)}|} \nabla \log[\overline{\pi}_\varphi(\boldsymbol{a}_t^{(w,j)}|\boldsymbol{s}_t^{(w,j)})] \Big)
$$
$$
- \prod_{i=1}^{\hat{N}_t} \frac{\pi_\theta(\hat{\boldsymbol{a}}_t^{(i)}|\hat{\boldsymbol{s}}_t^{(i)})}{\overline{\pi}_\varphi(\hat{\boldsymbol{a}}_t^{(i)}|\hat{\boldsymbol{s}}_t^{(i)})} \Big( \sum_{j=1}^{\hat{N}_t} \nabla \log[\overline{\pi}_\varphi(\hat{\boldsymbol{a}}_t^{(j)}|\hat{\boldsymbol{s}}_t^{(j)})] \Big) \Bigg);
$$
$$
-\nabla_\varphi \mu_l^S(\varphi, \theta) = -\sum_{t=1}^{T_l^S} \lambda_N(\boldsymbol{s}_t^{(l)}) \Bigg( \prod_{i=1}^{|A_t^{(l)}|} \frac{\pi_\theta(\boldsymbol{a}_t^{(l,i)}|\boldsymbol{s}_t^{(l,i)})}{\overline{\pi}_\varphi(\boldsymbol{a}_t^{(l,i)}|\boldsymbol{s}_t^{(l,i)})} \Big( \sum_{j=1}^{|A_t^{(l)}|} \nabla \log[\overline{\pi}_\varphi(\boldsymbol{a}_t^{(l,j)}|\boldsymbol{s}_t^{(l,j)})] \Big)
$$
$$
+ \prod_{i=1}^{\check{N}_t} \frac{\pi_\theta(\check{\boldsymbol{a}}_t^{(i)}|\check{\boldsymbol{s}}_t^{(i)})}{\overline{\pi}_\varphi(\check{\boldsymbol{a}}_t^{(i)}|\check{\boldsymbol{s}}_t^{(i)})} \Big( \sum_{j=1}^{\check{N}_t} \nabla \log[\overline{\pi}_\varphi(\check{\boldsymbol{a}}_t^{(j)}|\check{\boldsymbol{s}}_t^{(j)})] \Big) \Bigg),
\tag{43}
$$

where we also took the logarithmic scaling for the gradients in their implementation, *e.g.*,

$$
\prod_{i=1}^{|A_t^{(w)}|} \frac{\pi_\theta(\boldsymbol{a}_t^{(w,i)}|\boldsymbol{s}_t^{(w,i)})}{\overline{\pi}_\varphi(\boldsymbol{a}_t^{(w,i)}|\boldsymbol{s}_t^{(w,i)})} \rightarrow \prod_{i=1}^{|A_t^{(w)}|} \exp\big( \log(\pi_\theta(\boldsymbol{a}_t^{(w,i)}|\boldsymbol{s}_t^{(w,i)})) - \log(\overline{\pi}_\varphi(\boldsymbol{a}_t^{(w,i)}|\boldsymbol{s}_t^{(w,i)})) \big)
\tag{44}
$$

and so do others.

To train our policy networks in GSM8K, Game24, and MATH, we propose the preference data reconfigured from their training set. For each question $\boldsymbol{x}$ with correct response $\boldsymbol{y}$, we compute $-\log \pi_{\text{ref}}(\boldsymbol{y}'|\boldsymbol{x})$ across all responses $\boldsymbol{y}'$ from another

question to generate the hard negative preference pairs, then take them to initialize $\pi_\theta$ via cDPO. We set $\epsilon=0$ to human alignment task, yet set $\epsilon=0.25$ to mathematical reasoning and planning tasks. As for tree-based decoding, we employ the same tree-width pruning strategy in (Feng et al., 2024), difference only rises from the deterministic decision making or stochastic decision making. In GSM8K, we also provided the component analysis by varying $\epsilon$ in the range $\{100\%, 75\%, 50\%, 25\%, 0\%\}$, along with the results ranged from 53.2, 52.9, 53.4, 51.8, 50.1, accordingly. The performance drastically drops when the ratio less than 50%.

## D. LLM-based Reasoning Experiments Beyond Mathematics

We offer evaluation based on ProofWriter (Tafjord et al., 2020) for deductive logical reasoning, and Chess Endgame (Abdulhai et al., 2023) for long-term multi-turn decision making. For ProofWriter, we follow (Pan et al.) to generate the test set, then the rest are merged to 41,433 training instances. All training and test instances employed the prompt template in (Pan et al.) that initiated the start of CoT, then we employ LLAMA2-7B as the base model and all fine-tuning methods only run for a single epoch. For Chess Endgame, we follow the experimental setup in (Feng et al., 2023). With regards to each prompt-response pair $(\boldsymbol{x}, \boldsymbol{y}^{(w)})$, in ProofWriter and Chess Endgame, we find the dispreferred response $\boldsymbol{y}^{(l)}$ using the same strategy in our mathematical reasoning tasks. We ensure the evaluation in the fair comparison with the CoT and LLM-based tree-search baselines : CoT-greedy, BFS-V, MCTS- tr, MCTS-rollout, CoT-SC-MAJ, CoT-SC-ORM, BFS-V-ORM, MCTS-ORM, whose implementations are consistent in the paper.

For simplicity, we skip the average token number metric to highlight Acc in ProofWriter and Win rate in Chess Endgame. While their results remain based on Path@1 to promise the computation efficiency, and Equal-Token to encourage the comparison in the similar scale of computation consumption cross baselines. In the table, we found that CoT variants almost fail in ProofWriter due to their performances close to random guess (33.33%). MCTS variants obtain significantly better results yet basically under-perform ITS variants with substantial gap in ACC, probably due to the cold-start effect in MCTS learned with one epoch. As for Chess Endgame, ITS variants almost solve the problems with Win rates 99.83% in Path@1 and 98.57% in Equal-Token. It proves ITS also competitive in long-horizon reasoning.

