# OpenReview forum: "Language Models as Implicit Tree Search"
_ICML.cc/2025/Conference — ICML 2025 poster_

### Official Review · Reviewer_zco2 · 2025-03-01

**Overall Recommendation:** 3

**Summary:**

This submission proposes a novel preference optimization method for LLMs, grounded in the theoretical insight that LLMs can be viewed as implicit tree search, combining direct performance optimization with Monte Carlo Tree Search (MCTS).

**Claims And Evidence:**

No

**Essential References Not Discussed:**

No

**Experimental Designs Or Analyses:**

Experimental Designs Or Analyses are solid and extensive

**Methods And Evaluation Criteria:**

Yes

**Other Comments Or Suggestions:**

Line 137, a typo Convnetional

**Other Strengths And Weaknesses:**

Pros: This paper provides solid theoretical demonstration, connecting direct performance optimization and MCTS, showing a novel idea for  LLMs preferences alignment without additional reward or value model.

Cons: The concept of Implict Tree search is not clearly stated in the main part. The figure 2 in the appendix should be put in the main part, so that readers can have a clear high-level intuitive understanding at the begining. And authors should explain what the difference between explicit and implicit tree search and what the advantages of implicit tree search in the introduction seaction.

**Questions For Authors:**

What the difference between explicit and implicit tree search and what the advantages of implicit tree search? How to implement an  implicit tree search  process?

**Relation To Broader Scientific Literature:**

This submission is quiet related to reinforcement learning and MCTS,  like AlphaGo.

**Theoretical Claims:**

Yes, the proofs are correct

---

> ### Author Rebuttal · Authors · 2025-04-01
>
> Thanks for your comments and suggestions.
>
> **Q1. The concept of Implict Tree search (ITS) is not clearly stated in the main part, and authors should explain what the difference between explicit and implicit tree search and what the advantages of implicit tree search in the introduction section**
>
> **R1**: The concept ITS is derived from the theoretical result in (Grill et al. 2020), which verfiies that given each intermediate node in MCTS, if the intermediate node is treated as a state $s$ in RL, then the search process can be approximated by a state-specific stochastic policy $\overline{\pi}(\cdot|s)$ (Theorem.3.3). With such regards, for each AlphaZero-like MCTS, it always holds a stochastic variant defined by $\overline{\pi}(\cdot|s)$ across all possible intermediate nodes in the MCTS. However, $\overline{\pi}$ can not be universally optimized by gradient descent. Instead, given each state $s$, $\overline{\pi}(\cdot|s)$ can only be obtaind by solving a state-specific problem via dichotomic search. To this end, $\overline{\pi}(\cdot|s)$ can not be used as explicit search like MCTS and instead, only applicable for policy distillation to update the Q value function in AlphaZero-like MCTS (see Eq.7).
>
> The difference between MCTS (i.e., explicit tree search) and ITS mainly refers to three parts.
>
> First, MCTS is defined and executed by UCB in Eq.7, while ITS is defined through the set of policy optimization objectives (Eq.8) across all possible intermediate nodes in MCTS.
>
> Second, the execution of MCTS relies on the empirical visit distribution $\frac{1 + n(s,a)}{|\mathcal{A}{\sf MCTS}| + \sum_{a'\in \mathcal{A}_{\sf MCTS}}n(\boldsymbol{s},\boldsymbol{a}')}$. The empirical visit distribution suffers from the cold-start search problems and its expressivness is always lower than the continuous distrbituion. **It implies that ITS is more powerful than MCTS**.
>
> Third, ITS can not be explicitly used to search like MCTS due to its complexity to obtain the universal form of $\overline{\pi}$. Instead, ITS was mostly used in policy distillation to enhance Q-learning.
>
> The introduction would be updated to present more details of ITS.
>
> **Q2. The figure 2 in the appendix should be put in the main part, so that readers can have a clear high-level intuitive understanding at the begining; Line 137, a typo Convnetional**
>
> **R2**: Will rearrange the figures, and fix all typos in the next version.
>
> **Q3. How to implement an implicit tree search (ITS) process?**
>
> **R3**: In the original ITS defined in (Grill et al. 2020), it can not be implemented as the search strategy. However, in this paper, we establish a connection between ITS and DPO. So the universal ITS policy $\overline{\pi}$ can be approximated using a second LLM $\varphi$ by solving either Step-Synchronous IT-PO (Eq.16) or Step-Asynchronous IT-PO (Eq.18), alongside the LLM $\theta$ obtained through DPO-like algorithms. Once LLM $\varphi$ is properly trained, the ITS process can be implemented either by directly using $\varphi$ for next-token prediction or by employing our newly proposed decoding strategies: ITS-$\alpha$ and ITS-rollout.

---

> > ### Comment · Reviewer_zco2 · 2025-04-05
> >
> > Thanks for your responsible response that clarifies more details. I maintain my positive score.

---

### Official Review · Reviewer_REkf · 2025-03-13

**Overall Recommendation:** 4

**Summary:**

This paper introduces a novel preference optimization framework called Implicit Tree Search Preference Optimization (IT-PO) that addresses the reasoning limitations of existing LM alignment methods like Direct Preference Optimization (DPO). The key innovation is incorporating a second language model policy that approximates AlphaZero-like tree search behavior without requiring explicit tree construction or value modeling. By leveraging the asymptotic equivalence between stochastic policy optimization and MCTS, the authors formulate a preference optimization approach that maintains DPO's advantages while enabling more effective exploration and reasoning. The authors develop both step-synchronous and step-asynchronous variants of IT-PO, along with self-improved policy distillation and decoding strategies. Empirical evaluations demonstrate that IT-PO outperforms both DPO variants in human preference alignment tasks and surpasses MCTS-based language models in mathematical reasoning and planning benchmarks like GSM8K and Game24, while maintaining computational efficiency.

**Claims And Evidence:**

The paper's claims are generally well-supported by theoretical analysis and empirical evidence, though with some limitations. The mathematical formulation establishing asymptotic equivalence between implicit tree search and MCTS (Theorem 3.3) provides a sound theoretical foundation for the approach.

**Essential References Not Discussed:**

None

**Experimental Designs Or Analyses:**

I reviewed the experimental designs and analyses presented in the paper, focusing on the main empirical evaluations in Section 5.
The human preference alignment experiments (Section 5.1) use a standard methodology with the Anthropic HH dataset. The authors appropriately use Pythia 2.8B as the base model and employ consistent evaluation metrics (accuracy, diversity, win rate) across all methods. The win rate evaluation using GPT-4 follows established protocols in the field. The visualization of win rates across temperature scaling (Figure 1a) and across iteration rounds of policy distillation (Figure 1b) provides valuable insights into the model's behavior.
The mathematical reasoning experiments (Section 5.2) on GSM8K and Game24 datasets are well-designed. The authors properly implement both Path@1 and Equal-Token evaluation settings, allowing for fair comparison across computational budgets. Tables 2 and 3 effectively demonstrate performance across different search widths and node sizes, which helps validate the method's robustness.

**Methods And Evaluation Criteria:**

The proposed methods and evaluation criteria are appropriate and well-aligned with the paper's objectives. The authors use established datasets (Anthropic HH for preference alignment, GSM8K and Game24 for reasoning tasks) that are standard in the field, enabling fair comparison with baseline methods.

**Other Comments Or Suggestions:**

- The abbreviation "T-DPO" is used in Figure 1 but "TDPO" in the text, which creates confusion.

**Other Strengths And Weaknesses:**

**Strengths**
1. The authors establish a solid theoretical connection between Monte Carlo Tree Search (MCTS) and policy optimization through stochastic search, building on previous work by Grill et al. (2020). This provides a principled basis for their implicit tree search approach.
2. IT-PO elegantly combines the benefits of DPO (avoiding reward modeling) with the exploration capabilities of tree search algorithms, addressing a significant gap in current LM alignment methods.
3. Using two separate language models - one for token selection (πθ) and another for implicit tree search exploration (πφ) - is an innovative solution to the reasoning limitations of traditional preference optimization.

**Weaknesses**
1. While avoiding reward modeling, IT-PO introduces additional complexity by requiring two language models instead of one, which increases computational and memory requirements compared to standard DPO.
2. The experiments are conducted using relatively small models (Pythia 2.8B and Llama-7B). It remains unclear how IT-PO would scale to much larger language models, e.g. meta-llama/Llama-3.3-70B-Instruct.
3. While the GSM8K and Game24 benchmarks are standard, they represent relatively constrained reasoning tasks. Evaluation on more diverse and complex reasoning scenarios would strengthen the claims, e.g. MATH and AIME.

**Questions For Authors:**

1. Your method employs two language models (πθ and πφ) during inference, which potentially increases computational requirements compared to standard DPO. Could you provide quantitative metrics on the inference-time overhead of IT-PO compared to alternatives like standard DPO and MCTS-based approaches?
2. For the mathematical reasoning tasks, you mention generating hard negative preference pairs using responses from other questions. How sensitive is your method to the quality of these synthetic preference pairs?

**Relation To Broader Scientific Literature:**

The paper's key contributions intersect several important research directions in language model alignment, reinforcement learning, and reasoning capabilities.

**Theoretical Claims:**

I reviewed the theoretical claims and proofs presented in the paper, with particular focus on the foundational results in Sections 3 and 4.
The proof of Theorem 3.3, which establishes the asymptotic equivalence between implicit tree search policy and AlphaZero-like MCTS, appears to be mathematically sound. The authors clearly derive the upper bound on the approximation error between the empirical visit distribution and the stochastic policy solution. The theorem appropriately builds on prior work by Grill et al. (2020), and the adaptation to the language modeling context is logical.

Lemma 3.4, which provides the solution form for the implicit tree search policy, is correctly derived. The constraints and conditions for the parameter α(st) are well-specified, though the process of dichotomic search to find this parameter is somewhat glossed over in terms of practical implementation.

---

> ### Author Rebuttal · Authors · 2025-04-01
>
> Thanks for your comments and suggestions.
>
> **Q1. The limitation due to the demand of two language models.**
>
> **R1**: It is a good question since the requirement of LLM $\theta$ and $\varphi$ is inevitable in IT-PO. Despite so, the implementation can partially mitigate the limitation. First, despite the alternative training between $\theta$ and $\varphi$, we may only use one of them for inference, in particular, we may decode with either $\theta$ or $\varphi$ while the LLMs are derived from the IT-PO for token-level alignment. Second, since LLM $\varphi$ is the key to achieve reasoning, we may resort to the small-size LLM to reduce the training burden. The experiment of MATH in R2 verified this claim.
>
> **Q2. Larger language models as the base of IT-PO**;
>
> **Q3. Evaluation on more diverse and complex reasoning scenarios.**
>
> **R2-R3**: We provide the experiment to simultaneously address the concerns 2 and 3. It is because larger LLMs demand a more difficult task setup for the evaluation, and sufficiently complex reasoning task also requires a base model strong enough to support IT-PO. Specifically, we consider the reasoning-based evaluation on MATH and construct the training set integrated with the training splits of GSM8K and MATH. Due to our computation resource limited in the rebuttal period, comphrensive results based on 70B-size LLM are hardly achieved. Instead, we resort to other larger base LLMs in the Qwen family, i.e., Qwen1.5-32B.  Besides, we introduced LLAMA3.1-8B as a small LLM, aiming to verify our claim in R1. Then our evaluated LLMs ($\theta$, $\varphi$) in IT-PO are defined Qwen1.5-32BX2, (Qwen1.5-32B,LLAMA3-8B), then employed the decoding strategies ITS-$\alpha$ and ITS-rollout, respectively. For a comparison, we further train Qwen1.5-32B with MCTS to provide the baselines MCTS-$\alpha$ and MCTS-rollout, where the search depth = 4 and search width = 3. In our implementation, all IT-PO models are trained with the hyper-parameters consistent with GSM8K in SFT, cDPO, and IT-PO, while the search depth and breadth for training are not limited in MATH (i.e., LLM $\varphi$ trained as a stochastic policy without restriction). For decoding, all tree-search baselines use the search depth and width setup in GSM8K. The results are presented as
>
>  |  | Baselines/Decoding | greedy CoT | -$\alpha$ | -rollout |
> |--------|--------|--------|--------|--------|
> |  | Qwen1.5-32B | 36.1 | - | - |
> |  |MCTS- (Qwen1.5-32B)  | - | 36.0 | 36.7 |
> |  |ITS- ($\theta$=Qwen1.5-32B, $\varphi$=Qwen1.5-32B) | - | 39.8 | 40.2 |
> |  |ITS- ($\theta$=Qwen1.5-32B, $\varphi$=LLAMA3-8B) | - | 37.9 | 38.2 |
>
> In terms of the greedy CoT results in LLAMA3-8B (20.5), some observations can be found in the table. First, we note that the MCTS strategy derived from  is unreliable in MATH, probably due to the conflict between the problem complexity and limited search width and depth during training, while ITS does not suffer from this problem. Second, with a weaker model (LLAMA3-8B) for LLM $\varphi$, IT-PO still enable the improvement of LLM-based reasoning.
>
> **Q4. Confusion between T-DPO and TDPO.**
>
> **R4**: Will fix it in the next version.
>
> **Q5. Quantify the inference-time overhead of IT-PO compared to alternatives like standard DPO and MCTS-based approaches.**
>
> **R5**: The inference-time overhead of IT-PO can be significantly different w.r.t. decoding methods. For token-level alignment, IT-PO used a single model for decoding so that the overhead is consistent of standard DPOs. For reasoning tasks, it refers to ITS-$\alpha$ and ITS-$rollout$ strategies. ITS-$\alpha$ decodes the token like beam-search and for generated sentences, it evaluates their advantages via Eq.22. The process is similar with value inference, and ITS-$\alpha$ holds the similar inference time with MCTS-$\alpha$ (In GSM8K, each token is generated in $\sim$16ms by MCTS-$\alpha$ and in $\sim$18ms by ITS-$\alpha$ using a single A100 80G). ITS-$rollout$ can be treated as test-time-training version of ITS-$\alpha$, which updates $\varphi$ by Eq.21 to execute ITS-$\alpha$ afterwards. It generates 8 (K=8 in Reviewer MF7d's R1) responses for each test prompt to achieve self-training. In our experiment, it results in the extra 30min~1h for each evaluation.
>
> **Q6. The sensitivity of the preference pairs to the hard negative response.**
>
> **R6**: We evaluate ITS-$\alpha$ in GSM8K with hard negative ratio ranged from 100%, 75%, 50%, 25%, 0%, along with the results ranged from 53.2, 52.9, 53.4, 51.8, 50.1, accordingly. The performance drastically drops when the ratio less than 50%.

---

> > ### Comment · Reviewer_REkf · 2025-04-03
> >
> > Thank you for your thorough response. As my score reflects, I believe this is a commendable piece of work.

---

### Official Review · Reviewer_U4U8 · 2025-03-14

**Overall Recommendation:** 2

**Summary:**

The paper introduces a novel approach to preference optimization for LLM by incorporating Implicit Tree Search, drawing inspiration from Monte Carlo Tree Search and AlphaZero-like algorithms. The key contribution is an alternative preference optimization framework that allows LMs to implicitly execute a tree search strategy without relying on explicit value modeling, which is typically required in traditional MCTS-based methods.

**Claims And Evidence:**

The claims presented in the submission are partially supported by empirical evidence, specifically in the experimental comparisons provided in Tables 1-3, which demonstrate performance improvements in preference alignment and mathematical reasoning tasks. However, several claims remain problematic due to unclear presentations. Additionally, the assertion that the proposed method "inherits all advantages of DPO" without introducing drawbacks is unprecise. It's also concerning to use the language model policy as a "universal approximator".

**Essential References Not Discussed:**

N/A

**Experimental Designs Or Analyses:**

While empirical results mostly demonstrate some performance gains over existing methods, such improvements are often marginal rather than substantial. By the way, why are some numbers in table 2 missing?

**Methods And Evaluation Criteria:**

The proposed methods in the submission are conceptually relevant to the intended application—integrating implicit tree search (ITS) into language model preference optimization. The authors employ empirical comparisons using datasets (Anthropic HH, GSM8K, and Game24) suitable for evaluating preference alignment and reasoning capabilities. However, the performance improvements reported in these experiments are marginal in many cases, raising questions about the practical advantage and meaningfulness of the proposed methodological complexity.

**Other Comments Or Suggestions:**

N/A

**Other Strengths And Weaknesses:**

The manuscript currently lacks a clear motivation, intuition, and sufficient background prior to introducing complex mathematical content. Both the abstract and introduction do not effectively fulfill their intended roles of providing context and engaging readers.

Moreover, the manuscript includes many informal and imprecise statements, negatively impacting clarity and professionalism. Some specific examples from the abstract and introduction are:

- The phrase "... remains enabling LLM-based reasoning as AlphaZero"

- The sentence "While MCTS demands the value function to execute its procedure, contradicting the critical advantage of DPO that skips value learning to achieve faster and more stable preference optimization than RLHF," contains grammatical errors and requires restructuring for readability.

Additionally, the current title, "Language Models as Implicit Tree Search," is confusing. Based on the content presented in the main text, a more accurate title would be "Language Models as Implicit Tree Search Policies."

Addressing these points will enhance readability, precision, and overall scholarly quality.

**Questions For Authors:**

N/A

**Relation To Broader Scientific Literature:**

The key contributions of this paper build on prior literature by integrating ideas from Direct Preference Optimization (DPO), Reinforcement Learning from Human Feedback (RLHF), and Monte Carlo Tree Search (MCTS), particularly AlphaZero-like variants, into a unified implicit policy optimization framework. Specifically, the authors leverage theoretical results from Grill et al. (2020) that interpret AlphaZero-like MCTS as regularized policy optimization and propose using language models as implicit policies to replace explicit tree search.

**Theoretical Claims:**

The paper introduces a dense and highly formalized mathematical framework, but its presentation could be significantly improved. Currently, it lacks sufficient intuitive justification before diving into complex formulations, making it extremely difficult to follow.

I think it's a pity that the theoretical contribution is buried in dense formalism, making it hard to assess whether ITS is a fundamentally new insight or just a reformulation of existing search techniques.

---

> ### Author Rebuttal · Authors · 2025-03-31
>
> Thanks for your comments and concerns. We attempt to address the concerns by discussing the writing quality and the causes of marginal performance gain.
>
> **Q1: Writing quality.**
>
> **R1**: We response from three aspects.
>
> **First**, we appreciate the critique to phrases and grammar, and commit that our writing can be largely improved to encourage the readability. **In contrast to complicated terms, particularlly, our abstract and introduction, opt for more accessible and illustrative expressions to enhance comprehension**. Here are the argues to some cases found by Reviewer U4U8:
>
> Case 1.1 "... remains enabling LLM-based reasoning as AlphaZero": AlphaZero (Silver et, al. 2017b) is the famous Chess game Go system combining MCTS and value-based RL. Our IT-PO exactly uses the stochastic tree-search strategy derived from the consistent AlphaZero-like MCTS pipeline in (Feng, et al. 2023).
>
> Case 1.2 "It's also concerning... as a universal approximator": Policies separatly learned by state-specific objectives (Eq.8) can be uniformly approximated by the same LLM.
>
> Such writing style aims to provide a straightfoward sight of our work before diving into our formulations. It may also cause the unexpected confusion, which is promised to fix.
>
> **Second**,  though the clarity can be improved, we **do not agree** with the judge "unclear motivation and lack of sufficient background" to our work. Our motivation is intuitivly explained in the first paragraph: **LLM trained for preference alignment hardly achieves reasoning without value modeling or RL, in particular, LLM-based reasoning with MCTS.** To this, through analyzing the connection between MCTS and its stochastic variant (i.e. implicit tree search in our paper) , we propose **IT-PO, the first alignment framework that simultaneously achieve LLM-based reasoning without RL and value modeling**. Then we provide the prelimary and warm-up sections (2.5 pages) to elaborate our technical motivation and background. The comments in Reviewers MF7d, REkf, zco2, and also your summary, captured the motivation of this paper.
>
> Beyond this, we found that **some unjustified comment misinterpret our claim**, e.g.""inherits all advantages of DPO" without introducing drawbacks is unprecise" (we did not claim its escape from the DPO drawbacks.)
>
> **Third**, we did provide some intuitions to our theoretical parts. In Figure.2, we have illustrated the theoretical connection across MCTS, ITS derived from (Grill et, al. 2020b), and ITS learned by IT-PO. And each formula was always followed by its source and interpretation, e.g., 131-136 (right) for Theorem 3.3; 175-181 (left) for Lemma 3.4; 207-214(right), 234-237(left) for Lemma 4.1; 263-269 (left) for Theorem 4.2, etc. We commit that more illustrative examples can be introduced to improve the understanding of Theorem 4.3 (and will be refined), while the line of proofs in the paper still can be followed by the majority of reviewers.
>
> In general, we acknowledge the manuscript's writing deficiencies and promise for revision, yet we believe the aforementioned evidences insufficient to rate it below the ICML bar.
>
> **Q2: The causes of marginal performance gain.**
>
> **R2**: Our following experiments further explain, even refine the marginal gain:
>
> **Insufficient iterations of policy distillation**. In Sec.4.3, we know IT-PO trains LLM $\varphi$ along with cDPO with respect to fine-tuning LLM $\theta$, which constructs the alternative optimization between $\varphi$ and $\theta$. While in our implementation, we only set the iter=5 in AnthropicHH, and iter=1 in mathematical reasoning tasks (consistent with MCTS-based baselines). With regards to the analysis to Eq.19, increasing the iter number supposes to benefit the IT-PO performance. To this, we reset the IT-PO's evaluation with iter=7 in AnthropicHH, and iter=2 in GSM8K (we also reset iter=2 for MCTS for a fair comparison in Path@1). The results below support our claim.
>
> | AnthropicHH | Metric | TDPO$_2$ | IT-PO (ours, iter=5) | IT-PO (ours, iter=7) |
> |--------|--------|--------|--------|--------|
> |  | Alignment (Acc) | 67.33 | 69.12 | 70.83 |
> |  |Diversity (Ent)| 4.915 | 5.315 | 5.421  |
>
> | GSM8K | Baselines |  | iter=1 | iter=2 |
> |--------|--------|--------|--------|--------|
> |  | CoT-greedy |  | 41.4 | 41.4 |
> |  |BFS-V|  | 52.5 | 52.5  |
> |  |MCTS-$\alpha$  |  | 51.9 | 51.8 (-0.1) |
> |  |MCTS-rollout |  | 47.8 | 48.4 (+0.6) |
> |  |ITS-$\alpha$ (ours) |  | 53.2 | 54.3 (+1.1) |
> |  |ITS-rollout (ours) |  | 51.6 | 54.8 (+3.2) |
>
> **More appropriate experimental setup**. Based on the analysis in Sec.2, ITS could be promising in the cold-start search senarios, or more complex reasoning problems. This claim is supported by the results in ProofWriter (R2 in Reviewer MF7d, gain 6%$\sim$17%) and MATH (R2 in Reviewer REkf).
>
> **Q3. Missing numbers in table 2**
>
> **R3**: In GSM8K, the numbers in Equal-Token are consistent with those in Path@1 using tree-search baselines (see (Feng, et al. 2023)).

---

### Official Review · Reviewer_MF7d · 2025-03-18

**Overall Recommendation:** 3

**Summary:**

The paper proposes "Implicit Tree Search" (ITS), a stochastic approach that reformulates Monte Carlo Tree Search (MCTS) as a policy optimization problem. Unlike traditional MCTS methods that rely on discrete visit counts for exploration, ITS uses a continuous policy with reversed KL-divergence constraints.

**Claims And Evidence:**

The math checks out for their theoretical connection between ITS and MCTS - they've got formal proofs showing how their stochastic policy approach converges to AlphaZero-style search. And when they show their method beating DPO variants on that Anthropic human preference dataset, the numbers in their tables and graphs seem convincing. They're also showing competitive results on those math reasoning benchmarks compared to MCTS approaches.

**Essential References Not Discussed:**

I think most of the citations are discussed in the paper.

**Experimental Designs Or Analyses:**

I examined several aspects of the experimental designs and analyses in the paper. The key experiments involve comparing IT-PO against baseline methods on preference alignment tasks (Anthropic HH dataset) and mathematical reasoning tasks (GSM8K and Game24).
For the preference alignment experiments (Section 5.1), I looked at their design for evaluating accuracy, diversity, and win rates. The way they measure these metrics is relatively standard in the field. They use accuracy on chosen vs. rejected completions, predictive entropy for diversity, and GPT-4 judgments for win rates - all common approaches.
Their win rate evaluation method involves a temperature scaling study (Figure 1a) which shows how performance varies with different sampling temperatures. This is a solid approach to examine robustness across different decoding strategies. The preference policy distillation experiments (Figure 1b) also make sense methodologically - they're measuring improvement over iterations of their approach.
For the mathematical reasoning experiments (Section 5.2), I examined their Path@1 and Equal-Token evaluation setups. The Path@1 metric (success rate for first-path generation) makes sense for evaluating reasoning quality. The Equal-Token comparison is a good attempt to control for computational budget differences between methods.

**Methods And Evaluation Criteria:**

Their Implicit Tree Search approach tackles a real problem with traditional MCTS - those discrete visit counts can be limiting, especially early in the search process. Moving to a stochastic policy feels like a natural solution. And I like how they've connected this to preference optimization without needing explicit rewards - that's clever and practical given recent trends.
The datasets they picked seem reasonable. Anthropic HH is pretty standard for testing human preference stuff. And those math datasets (GSM8K and Game24) make sense since they need multi-step reasoning where tree search should help.

**Other Comments Or Suggestions:**

I don't have any major comment left.

**Other Strengths And Weaknesses:**

Weakness: While the mathematical reasoning tasks do demonstrate reasoning capabilities, testing on a wider range of reasoning domains (logical puzzles, programming tasks, etc.) would strengthen generalizability claims.

**Questions For Authors:**

Your evaluation focuses primarily on mathematical reasoning tasks and preference alignment. Have you tested IT-PO on other reasoning-heavy domains such as logical puzzles, programming tasks, or complex planning scenarios? Evidence of broader applicability would strengthen your claim that this approach provides general reasoning improvements

**Relation To Broader Scientific Literature:**

The paper effectively creates a bridge between two previously distinct approaches: direct preference optimization methods that are computationally efficient but struggle with complex reasoning, and tree search methods that excel at reasoning but require explicit reward modeling and significant computational resources. By reformulating tree search as implicit policy optimization, it offers a novel way to get "reasoning with free lunch" - complex reasoning capabilities without the overhead of explicit tree search or reward modeling.

**Theoretical Claims:**

I took a look at several of the theoretical claims and proofs in the paper. The main ones I examined were Theorem 3.3 about the asymptotic equivalence between ITS and MCTS policies, Lemma 3.4 on the solution of ITS policy, and the proofs surrounding Lemma 4.1 and Theorem 4.2 that establish the connection between preference optimization and their implicit tree search approach.
Overall the proofs seem technically correct for the most part, but they sometimes gloss over implementation details that would matter in practice. The theoretical foundation is sound enough to justify their approach, even if some of the finer points might benefit from more detailed explanation or justification.

---

> ### Author Rebuttal · Authors · 2025-03-31
>
> Thanks for your comments and suggestions.
>
> **Q1. More implementation details behind theory ("they sometimes gloss...more detailed explanation").**
>
> **R1**: Despite Theorems 4.2, 4.3 containing numerous variables, most are derived from the data rather than serving as hyperparameters. Thus, IT-PO holds implementation consistency with its theoretical foundation except for only two details:
>
> 1.1 $\lambda_N(s^{(w)}_t),\lambda_N(s^{(l)}_t)$ indicate how many reasoning paths generated from the state $s_t$, which consists of a prompt $x$ and its responses $y^{(w)}$,$y^{(l)}$ with their contents in the previous $t-1$ steps. For each preference pair, we set $\forall t\in${$1,\cdots,T_w$}, $\lambda_N(s^{(w)}_t)=K_w$ and $\forall t\in${$1,\cdots,T_l$}, $\lambda_N(s^{(l)}_t)=K_l$, where $K_w$, $K_l$ denote how many search start from the $t$-th leaf nodes $s^{(w)}_t, s^{(l)}_t$, respectively. Note that $T_w, T_l$ change with respect to the preference pair. So IT-PO adaptively configure $K_w=\frac{K}{T_w}$, $K_l=\frac{K}{T_l}$ to balance the optimization with different response lengths in $y^{(w)}$,$y^{(l)}$ for each pair. We set $K=8$, inspired from the number of sampled responses for each prompt in many RLHF implementations.
>
> 1.2 In gradient analysis in Eq.19, the gradient of $\varphi$ consists of terms in the form $\frac{\pi{\theta}(a_t|s_t)}{\overline{\pi}{\varphi}(a_t|s_t)}$ $\nabla \log [{\overline{\pi}}(a_t|s_t)]$. Due to $\frac{\pi{\theta}(a_t|s_t)}{\pi{\varphi}(a_t|s_t)}\in(0,+\infty)$, updating the models with $\frac{\pi{\theta}(a_t|s_t)}{\overline{\pi}{\varphi}(a_t|s_t)}$ $\nabla \log [{\overline{\pi}}(a_t|s_t)]$ may suffer from exploding/vanishing gradients. To this, we used their logarithmic scaling $\frac{\pi{\theta}(a_t|s_t)}{\overline{\pi}{\varphi}(a_t|s_t)}$$\rightarrow$ $\exp(\log(\pi{\theta}(a_t|s_t))-\log(\overline{\pi}{\varphi}(a_t|s_t)))$ to ensure the less sensitive update ratio. We also took the logarithmic scaling for the gradients in the Step-Asynchronous case (Eq.17), which will be elaborated in our next version due to our rebuttal limit.
>
> **Q2. More experiments for other reasoning tasks (Weakness and Questions For Authors).**
>
> **R2**: We offer evaluation based on ProofWriter [1] for deductive logical reasoning, and Chess Endgame [2] for long-term decision making. For ProofWriter, we follow [3] to generate the test set, then the rest are merged to 41,433 training instances. All training and test instances employed the prompt template in [3] that initiated the start of CoT, then we employ LLAMA2-7B as the base model and all fine-tuning methods only run for a single epoch. For Chess Endgame, we follow the experimental setup in (Feng, et al. 2023). With regards to each prompt-response pair $(x,y^{(w)})$ in ProofWriter and Chess Endgame, we find the dispreferred response $y^{(l)}$ using the same strategy in our mathematical reasoning tasks. We ensure the evaluation in the fair comparison with the CoT and LLM-based tree-search baselines : CoT-greedy, BFS-V, MCTS-$\alpha$tr, MCTS-rollout, CoT-SC-MAJ, CoT-SC-ORM, BFS-V-ORM, MCTS-ORM, whose implementations are consistent in the paper.
>
> | Setting | Baselines |  | ProofWriter (Acc %) | Chess Endgame (Win rate %) |
> |--------|--------|--------|--------|--------|
> | Path@1 | CoT-greedy |  | 37.72 | 58.14 |
> |  |BFS-V|  | 48.94 | 67.75  |
> |  |MCTS-$\alpha$  |  | 66.71 | 96.90 |
> |  |MCTS-rollout |  | 69.23 | 98.76 |
> |  |ITS-$\alpha$ (ours) |  | 71.77 | 99.21 |
> |  |ITS-rollout (ours) |  | 75.31 | 99.83 |
> |--------|--------|--------|--------|--------|
> |Equal-Token  |CoT-SC-MAJ|  | 36.50 | 9.84 |
> |  |CoT-SC-MAJ|  | 36.58 | 73.80 |
> |  |BFS-V-ORM|  | 63.42 | 93.18 |
> |  |MCTS-ORM|  | 60.86 | 94.26 |
> |  |ITS-$\alpha$ (ours)|  | 74.26 | 96.48 |
> |  |ITS-rollout (ours)|  | 78.15 | 98.57 |
>
> For simplicity, we skip the average token number metric to highlight Acc in ProofWriter and Win rate in Chess Endgame. While their results remain based on Path@1 to promise the computation efficiency, and Equal-Token to encourage the comparison in the similar scale of computation consumption cross baselines. In the table, we found that CoT variants almost fail in ProofWriter due to their performances close to random guess (33.33%). MCTS variants obtain significantly better results yet basically under-perform ITS variants with substantial gap in ACC, probably due to the cold-start effect in MCTS learned with one epoch. As for Chess Endgame, ITS variants almost solve the problems with Win rates 99.83% in Path@1 and 98.57% in Equal-Token. It proves ITS also competitive in long-horizon reasoning.
>
> Ref:
>
> 1. Tafjord, O., et al. ProofWriter: Generating Implications, Proofs, and Abductive Statements over Natural Language (ACL-IJCNLP 2021).
>
> 2. Abdulhai, M., et al. Lmrl gym: Benchmarks for multi-turn reinforcement learning with language models. arXiv preprint
>
> 3. Pan L, et al. Logic-LM: Empowering Large Language Models with Symbolic Solvers for Faithful Logical Reasoning[C] EMNLP-2023

---

### Decision · Program_Chairs · 2025-05-01

**Decision:**

Accept (poster)

**Comment:**

This paper provides a theoretical connection between reward-guided text generation (e.g. under (tokenwise) alignment algorithms like DPO) and AlphaZero-like MCTS as a regularized policy optimization (Grill et al., 2020). Based on this insight, the authors proposed a method called Implicit Tree Preference Optimization (IT-PO), which uses an additional language model to approximate the AlphaZero policy. This additional policy network is trained via a tokenwise loss similar to DPO. Experiments show that IT-PO can achieve better alignment results than various baselines.

The majority of the reviewers agree that the proposed analysis and method are novel and interesting. I agree with this assessment.

Nevertheless, I also agree with the reviewers' assessments that the presentation of the paper is its weakest point. There are many grammatical errors and typos in the manuscript. The exposition of the main idea of the paper and the method are also quite confusing. I suggest the authors move Fig. 2 in the appendix to the main text and package the method into a dedicated algorithm environment.